# Fine-mapping analysis including over 254,000 East Asian and European descendants identifies 136 putative colorectal cancer susceptibility genes

Genome-wide association studies (GWAS) have identified more than 200 common genetic variants independently associated with colorectal cancer (CRC) risk, but the causal variants and target genes are mostly unknown. We sought to fine-map all known CRC risk loci using GWAS data from 100,204 cases and 154,587 controls of East Asian and European ancestry. Our stepwise conditional analyses revealed 238 independent association signals of CRC risk, each with a set of credible causal variants (CCVs), of which 28 signals had a single CCV. Our cis-eQTL/mQTL and colocalization analyses using colorectal tissue-specific transcriptome and methylome data separately from 1299 and 321 individuals, along with functional genomic investigation, uncovered 136 putative CRC susceptibility genes, including 56 genes not previously reported. Analyses of single-cell RNA-seq data from colorectal tissues revealed 17 putative CRC susceptibility genes with distinct expression patterns in specific cell types. Analyses of whole exome sequencing data provided additional support for several target genes identified in this study as CRC susceptibility genes. Enrichment analyses of the 136 genes uncover pathways not previously linked to CRC risk. Our study substantially expanded association signals for CRC and provided additional insight into the biological mechanisms underlying CRC development.

Colorectal cancer (CRC) is one of the most common malignancies worldwide[1]. Inherited genetic factors play an important role in the development of CRC[2]. Since 2007, genome-wide association studies (GWAS) have identified over 200 common genetic variants independently associated with CRC risk[3–7]. These GWAS, however, typically only reported the most significantly associated variant (the lead variant) at each risk locus. Statistical fine-mapping analyses of known risk loci can identify additional association signals independent of the lead variant.

Approximately 90% of GWAS-identified risk variants for CRC are located in noncoding or intergenic regions, and target genes for most of these risk variants remain unknown. Well-powered fine-mapping analyses, particularly those using data from multi-ancestry populations, can facilitate the identification of credible causal variants (CCVs) in each region. Previous genetic studies have provided strong evidence that regulatory variants in linkage disequilibrium (LD) with GWAS-identified risk variants drive the associations of genetic variants with cancer risk by modulating the expression of susceptibility genes[8–11]. Therefore, integrating functional genomic data to interrogate CCVs in each independent risk-associated signal could help to identify putative causal variants and target genes for CRC risk. Herein, we conducted large trans-ancestry fine-mapping analyses of all currently known CRC

e-mail: wei.zheng@vanderbilt.edu

risk regions, using GWAS data from 100,204 CRC cases and 154,587 controls of East Asian and European ancestry, to identify independent association signals and their target genes for CRC risk.

## Results

### Identification of independent association signals with CRC risk

We conducted fine-mapping analyses using GWAS summary statistics from 100,204 CRC cases and 154,587 controls (73% European and 27% East Asian ancestry) (Fig. 1, Supplementary Data 1). In our recent trans-ancestry meta-analysis of GWAS, we identified 205 genetic variants independently associated with CRC risk[7]. We aggregated regions flagged by these variants into 143 risk regions, each containing at least a 1 Mb interval centered on the most significant association (Supplementary Data 2). Among them, 40 regions harbor at least two reported independent risk associations. All risk regions were autosomal, except the one at Xp22.2. For subsequent analyses, we focused on the 142 regions located on the autosomes.

We used forward stepwise conditional analyses to identify independent association signals in each region in each population, conditioning on the most significant association from the trans-ancestral summary statistics (Supplementary Fig. 1, Methods). We then meta-analyzed the conditioned data using the fixed-effects inverse variance weighted model. We considered the threshold of conditional $P < 1 \times 10^{-6}$ to determine independent significant associations to balance both Type 1 and 2 errors, as recommended by a previous fine-mapping study in breast cancer[12]. At this threshold, we identified 171 independent association signals in 122 regions (Fig. 2, Supplementary Data 3). To identify possible ancestry-specific association signals, we conducted similar analyses using only summary statistics from each

population, conditioning on the ancestry-specific most significant association. Using the same threshold, we identified 198 and 45 independent association signals in European and East Asian descendants, respectively (Supplementary Data 4 and 5). Of them, 60 signals in European and 7 in East Asian were not detected in the trans-ancestry analysis above, suggesting them as potential ancestry-specific risk signals (Fig. 2).

In total, we identified 238 independent association signals either from trans-ancestry or ancestry-specific analysis at these 142 regions (Fig. 2). A total of 94 regions (66.2%) contained only a single association signal, while the remaining 48 regions (33.8%) consisted of multiple independent association signals. Among the 238 independent association signals, 191 signals had lead variants that were correlated with previously GWAS-reported risk variants[7] (LD $r^2 > 0.1$ in either of East Asian or European-ancestry population). The remaining 47 independent signals (19.7%) have not been previously reported, including 18 from trans-ancestry, 28 from European-specific, and one from East Asian-specific analyses (Fig. 2, Table 1). Among these 47 signals, 31 demonstrated significant associations with conditional $P < 1 \times 10^{-7}$, including 28 signals reached genome-wide significance.

### Identification of credible causal variants (CCVs) for independent association signals

To identify CCVs for each independent association signal, we conducted conditional analysis with adjustment of the lead variants for other signals in the same risk region. We conducted this analysis for trans-ancestral independent signals separately for each population to account for differences in the LD structure and then meta-analyzed conditioned results. Using a similar approach conducted in breast cancer[12], we defined variants as CCVs if they satisfied conditional $P$ values within two orders of magnitude of the most significant association, conditioning on all other independent association signals. We identified a total of 5741 CCVs for the 238 signals, with the number of CCVs per signal ranging from 1 to 249 (median: 11 CCVs per signal) (Supplementary Data 6). For 28 risk signals, only a single CCV was identified, suggesting that these CCVs are likely to be the causal variants for these signals (Table 2).

For the 138 independent association signals identified in both trans-ancestry and European-ancestry specific analyses (Supplementary Data 7), trans-ancestry analyses identified a smaller-sized set of CCVs (mean = 23.2, median = 8.5), compared with European-ancestry specific analysis (mean = 31.08, median = 15) (paired Wilcoxon test, $P = 4.9 \times 10^{-7}$). Interestingly, a single CCV was identified for 10 signals in trans-ancestry analysis, while multiple CCV for them in European-ancestry specific analysis, highlighting the value of using multi-ancestry data to reduce the number of CCVs in fine-mapping analysis. For instance, signal 1 in region_42 included 16 CCVs in the European set (lead variant: rs41302867), but only one variant in the trans-ancestry set (rs9379084). The variant rs9379084 is a predicted-deleterious missense variant (p.Asp1171Asn) of the *RREB1* gene which plays a regulatory role in Ras/Raf-mediated cell differentiation[13], a pathway well known to be implicated in CRC development.

### Identification of target genes for CCVs

Of the 5741 CCVs identified in this study, 3716 (64.7%) are located in regions with at least one of six genomic features (open chromatin, transcribed regions of active genes, promoter, enhancer, repressed gene regulatory elements, and transcription factor (TF) binding sites) (Supplementary Data 6 and 8). To identify putative target genes of these CCVs, we used functional genomic data generated in CRC-related tissues/cells to conduct in-silico analyses with a modified INQUISIT pipeline[12] (Methods, Supplementary Data 9). We identified 72 putative target genes via CCVs located in distal enhancer elements (Supplementary Data 10), 48 genes via CCVs located in proximal promoter elements (Supplementary Data 11), and 19 genes that could be

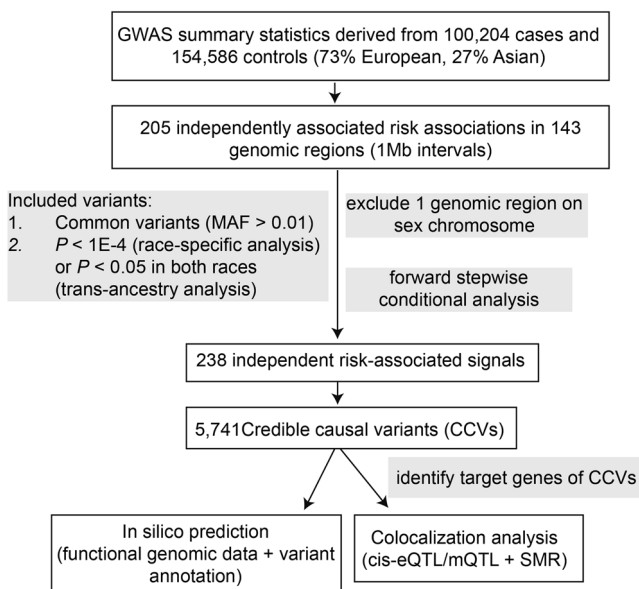

**Fig. 1 | Schematic diagram of the study design.** We conducted fine-mapping analyses using GWAS summary statistics from 100,204 cases and 154,587 controls. All 205 genetic variants were aggregated to 143 risk regions containing at least a 1 megabase (Mb) interval centered on the most significant association. This study focused on 142 risk regions located on the autosomes. In forward stepwise conditional analysis, we included common variants (minor allele frequency (MAF) > 0.01) with associations at $P < 0.05$ in both populations for the trans-ancestry analysis and with associations at $P < 1 \times 10^{-4}$ in each population for race-specific analysis. The threshold of conditional $P < 1 \times 10^{-6}$ was used to determine independent risk-associated signals. For credible causal variants (CCVs) for each independent signal, we conducted *in-silico* analyses with functional genomic data generated in CRC-related tissues/cells and colocalization of expression/methylation quantitative trait loci (e/mQTL) with GWAS signals to identify putative target genes for CCVs using the Summary-data-based Mendelian Randomization (SMR) approach.

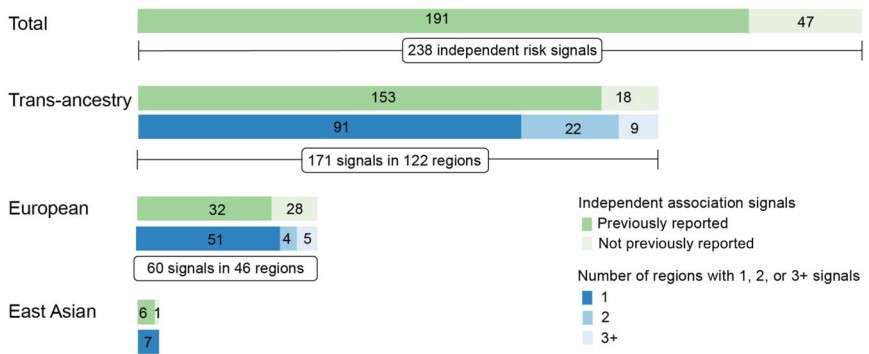

**Fig. 2 | Independent association signals for colorectal cancer risk.** Numbers of fine-mapping regions and numbers of independent association signals identified through forward stepwise conditional analyses. The second bar for "Trans-ancestry", "European" and "East Asian" also shows the number of regions with 1, 2, or 3+ signals per region. The green color indicates the number of independent association signals previously reported or not yet reported. The blue color indicates the number of independent association signals in each risk region.

targeted by CCVs in coding regions (i.e., deleterious missense, stop_gained, and start_lost) (Supplementary Data 12). In total, we identified 128 genes associated with CCVs for 76 independent association signals, with a range from one to five putative target genes per signal. Of them, 52 independent association signals contain only a single putative target gene.

We also conducted cis-expression quantitative trait loci (cis-eQTL) analyses to identify target genes using four transcriptome datasets derived from either normal colon tissues or tumor-adjacent normal colon tissues from 1299 individuals from the Genotype-Tissue Expression (GTEx) project ($n = 368$ individuals predominantly of European ancestry), the BarcUVa-Seq project ($n = 423$ individuals of European ancestry), the Colonomics project ($n = 144$ individuals of European ancestry), and the Asia Colorectal Cancer Consortium (ACCC) ($n = 364$ individuals of East Asian ancestry) (Methods). At Bonferroni-corrected $P < 0.05$, we identified 153 genes associated with the lead variants, including 127 genes in 65 independent association signals and 30 in 15 signals identified from trans-ancestry and European-ancestry specific analyses, respectively. We also identified the *PPP1R21* gene in a potential Asian-specific risk signal (lead variant rs77272589) (Supplementary Data 13). Out of the 153 genes, 37 had been previously identified by eQTL analysis[5,10,11]. For independent association signals identified in European and trans-ancestry analyses, we further performed cis-methylation quantitative trait loci (cis-mQTL) analyses using two methylation datasets generated from 321 individuals from the GTEx project ($n = 189$ individuals predominantly of European ancestry) and the Colonomics project ($n = 132$ individuals of European ancestry). We found that DNA methylation levels at CpG sites for 84 genes were associated with 71 independent association signals, including 14 genes identified in previous mQTL analysis[11] (Supplementary Data 14).

We next conducted colocalization analyses for identified likely target genes in significant eQTL/mQTLs above using the Summary-data-based Mendelian Randomization (SMR) approach (Methods). Through the integration of eQTL/mQTL results and GWAS associations signals, we identified 205 genes at Bonferroni-corrected $P_{SMR} < 0.05$ (Supplementary Data 15–19), including 150 genes from the eQTL analysis and 84 genes from the mQTL analysis. Of these, 45 (21.9%) genes were also identified as targets of CCVs by in-silico analyses based on functional genomic data as described above, and 29 genes were identified in both mQTL and eQTL analyses. That is in line with previous observations in the overlap fraction between mQTL and eQTL[14]. We considered genes with evidence of only mQTL colocalization, as the enrichment of mQTLs in gene regulatory elements, as well as their implications in other molecular phenotypes, such as chromatin accessibility[14,15]. Notably, of the 55 genes only identified in the mQTL

analysis, seven genes were supported by the above in silico analyses with functional genomic data, and 22 genes showed association with CRC risk in previous TWAS and eQTL colocalization analysis[7,11,16,17].

In total, we identified 288 putative target genes for 140 independent association signals based on functional genomics data and/or colocalization analysis. For 35 of these signals, multiple target gene candidates were detected per signal, suggesting that some may be false positives (Supplementary Data 20). To minimize false positive findings, we further prioritized target gene candidates by analyzing associations of genes with CRC risk based on previous transcriptome-wide association studies (TWAS) and colocalizations between eQTL and CRC GWAS signals[7,11,16,17] (Methods). Finally, we obtained a credible set of 136 protein-coding genes for 124 independent association signals. Among them, 56 genes were not previously identified as potential targets for CRC risk associations, including nine genes in eight previously unreported association signals in this study (Table 3). The remaining 80 genes were previously reported as potential CRC susceptibility genes, and our study provided additional supporting evidence (Table 4)[7,11,16,17].

## Using scRNA-seq data to evaluate gene expression pattern by cell types

To investigate potential underlying cell types of putative susceptibility genes that contribute to CRC development, we analyzed single-cell RNA-seq (scRNA-seq) datasets from normal colon tissues obtained from 31 participants included in the Colorectal Molecular Atlas Project[18] (Methods). Of the 136 identified genes, 17 genes exhibited significantly differential expression in specific cell types compared to the other cell types at |log2 fold change (FC)| > 1 and a nominal $P < 0.05$ (Supplementary Data 21). Nine of these genes (*DIP2B*, *CIB1*, *HPGD*, *CDKN2B*, *TMEM258*, *MYL12A*, *MYL12B*, *CDKN1A*, and *TMBIM1*) showed a distinct expression pattern in specific absorptive cells (ABS) cell, underscoring the relevance of this cell type underlying CRC development.

## Using whole exome sequencing data to evaluate pathogenic variants in target genes with CRC risk

We used whole exome sequencing data from 3362 CRC cases and 133,742 controls of European ancestry in the UK Biobank (UKBB) to evaluate the association of CRC risk with putative candidate genes identified our study using burden tests by aggregating either loss of function (pLOF) or pLOF and deleterious missense variants (Dmis) jointly in each gene (Methods). Of these 136 genes, *MLH1* was significantly associated with CRC risk with $P = 1.35 \times 10^{-7}$ when considering only pLOF in tests (at Bonferroni-corrected threshold, 0.05/136 testing). Additional nine genes (*TNFSF18*, *LRP1*, *SMAD9*, *PDGFB*, *CIB1*,

**Table 1 | Independent association signals uncovered at known CRC risk loci in conditional analyses (conditional $P < 1 \times 10^{-6}$)**

| Fine-mapping region | SNP | Chr | Position | Nearby gene | Alleles | AF | Single-SNP analysis | | Joint analysis | | Group |
|---|---|---|---|---|---|---|---|---|---|---|---|
| | | | | | | | OR (95% CI) | P value[a] | OR (95% CI) | P value[b] | |
| region_1 | rs115579545 | 1 | 22249333 | HSPG2 | T/C | 0.445 | 0.96 (0.95–0.98) | 4.34E-07 | 0.96 (0.95–0.98) | 5.63E-07 | Trans-ancestry |
| region_1 | rs112191583 | 1 | 22554378 | MIR4418 | T/C | 0.974 | 0.88 (0.83–0.92) | 1.19E-07 | 0.87 (0.83–0.92) | 5.29E-08 | Trans-ancestry |
| region_1 | rs12137525 | 1 | 2584118 | MIR4418 | T/C | 0.107 | 1.07 (1.04–1.09) | 2.90E-08 | 1.08 (1.06–1.11) | 1.14E-11 | European |
| region_9 | rs12122827 | 1 | 202172769 | LGR6 | T/G | 0.715 | 1.04 (1.02–1.06) | 9.44E-07 | 1.05 (1.03–1.06) | 7.94E-08 | European |
| region_22 | rs2554878 | 3 | 41200064 | RP11-372H2.1 | T/G | 0.036 | 1.12 (1.08–1.16) | 5.85E-09 | 1.12 (1.08–1.16) | 3.75E-09 | Trans-ancestry |
| region_27 | rs9283588 | 3 | 133874566 | RYK | A/G | 0.715 | 1.06 (1.04–1.07) | 3.43E-10 | 1.04 (1.03–1.06) | 7.32E-07 | Trans-ancestry |
| region_30 | rs902443 | 4 | 105888417 | RP11-556I14.1 | A/T | 0.536 | 1.04 (1.03–1.06) | 1.49E-11 | 1.04 (1.03–1.06) | 1.26E-11 | Trans-ancestry |
| region_36 | rs582489 | 5 | 39908712 | GCSHP1 | T/C | 0.570 | 0.97 (0.96–0.99) | 8.23E-05 | 0.96 (0.94–0.97) | 7.29E-09 | European |
| region_36 | rs77781678 | 5 | 40626064 | SNORA63 | T/C | 0.020 | 0.84 (0.79–0.89) | 2.09E-09 | 0.84 (0.79–0.89) | 1.75E-09 | European |
| region_43 | rs4714081 | 6 | 11977905 | RP11-456H18.1 | A/G | 0.451 | 0.96 (0.95–0.97) | 2.50E-09 | 0.96 (0.95–0.97) | 1.21E-09 | Trans-ancestry |
| region_43 | rs4714350 | 6 | 12270290 | EDN1 | A/T | 0.283 | 0.96 (0.94–0.97) | 8.60E-09 | 0.96 (0.95–0.98) | 4.43E-07 | Trans-ancestry |
| region_43 | rs17615624 | 6 | 12376025 | RN7SKP293 | C/G | 0.975 | 0.87 (0.83–0.91) | 7.29E-09 | 0.88 (0.84–0.92) | 2.28E-07 | European |
| region_44 | rs3094576 | 6 | 29516242 | OR2I1P | A/C | 0.131 | 0.94 (0.92–0.96) | 1.83E-07 | 0.94 (0.92–0.96) | 2.26E-08 | European |
| region_44 | rs2517671 | 6 | 29937977 | MICD | A/G | 0.591 | 0.96 (0.95–0.98) | 2.35E-08 | 0.96 (0.95–0.97) | 2.87E-09 | European |
| region_45 | rs6920820 | 6 | 30969938 | MUC22 | C/G | 0.980 | 0.84 (0.79–0.9) | 6.87E-08 | 0.8 (0.75–0.85) | 1.89E-12 | European |
| region_45 | rs9264180 | 6 | 31219902 | HLA-C | A/C | 0.570 | 1.03 (1.02–1.05) | 1.71E-06 | 1.04 (1.02–1.05) | 5.62E-07 | Trans-ancestry |
| region_45 | rs9265501 | 6 | 31297568 | XXbac-BPG248L24.10 | A/G | 0.678 | 0.88 (0.85–0.92) | 3.05E-10 | 0.88 (0.84–0.91) | 5.21E-11 | European |
| region_45 | rs116000952 | 6 | 32541270 | HLA-DRB1 | T/G | 0.843 | 0.92 (0.89–0.96) | 5.74E-06 | 0.9 (0.87–0.94) | 1.50E-08 | Trans-ancestry |
| region_45 | rs2858331 | 6 | 32681277 | XXbac-BPG254F23.7 | A/G | 0.601 | 1.03 (1.02–1.05) | 1.18E-05 | 1.05 (1.04–1.07) | 2.67E-13 | Trans-ancestry |
| region_50 | rs13204733 | 6 | 55566108 | RP11-228O6.2 | A/G | 0.858 | 0.94 (0.92–0.96) | 4.20E-08 | 0.93 (0.91–0.95) | 1.17E-12 | European |
| region_60 | rs10089517 | 8 | 60178721 | SNORA51 | A/C | 0.380 | 1.03 (1.02–1.05) | 7.44E-07 | 1.03 (1.02–1.05) | 2.81E-07 | European |
| region_61 | rs117310502 | 8 | 117593052 | EIF3H | A/G | 0.048 | 0.92 (0.89–0.96) | 9.36E-05 | 0.88 (0.85–0.92) | 4.03E-10 | European |
| region_61 | rs72681666 | 8 | 117641754 | EIF3H | T/C | 0.043 | 1.09 (1.05–1.13) | 1.57E-05 | 1.12 (1.08–1.17) | 6.99E-10 | European |
| region_61 | rs1793717 | 8 | 118278575 | SNORA31 | A/C | 0.629 | 1.03 (1.02–1.05) | 6.90E-05 | 1.04 (1.03–1.06) | 1.55E-07 | European |
| region_62 | rs79122086 | 8 | 128397907 | CASC8 | T/G | 0.840 | 0.92 (0.9–0.93) | 5.46E-20 | 0.94 (0.93–0.96) | 9.34E-10 | Trans-ancestry |
| region_62 | rs77569096 | 8 | 128468955 | CASC8 | A/G | 0.763 | 0.92 (0.9–0.94) | 2.06E-15 | 0.93 (0.91–0.95) | 2.67E-12 | European |
| region_68 | rs4994332 | 9 | 137117194 | RP11-145E17.2 | T/C | 0.423 | 0.97 (0.96–0.98) | 4.05E-05 | 0.96 (0.95–0.97) | 9.08E-08 | European |
| region_74 | rs117746067 | 10 | 101222300 | RP11-441O15.3 | A/G | 0.101 | 1.06 (1.03–1.08) | 3.64E-06 | 1.08 (1.05–1.1) | 1.74E-09 | Trans-ancestry |
| region_80 | rs9795065 | 11 | 74376844 | POLD3 | T/C | 0.981 | 1.19 (1.13–1.25) | 5.37E-13 | 1.17 (1.12–1.23) | 6.06E-11 | Trans-ancestry |
| region_85 | rs1003563 | 12 | 6424577 | PLEKHG6 | A/G | 0.433 | 0.95 (0.94–0.97) | 7.52E-06 | 0.95 (0.94–0.96) | 1.23E-14 | Trans-ancestry |
| region_106 | rs68097734 | 14 | 92717447 | RP11-472N19.3 | T/C | 0.496 | 1.06 (1.03–1.09) | 7.71E-06 | NA | – | Asian |
| region_108 | rs28630996 | 15 | 32993860 | SCG5 | A/T | 0.713 | 0.9 (0.89–0.92) | 1.25E-32 | 0.93 (0.91–0.94) | 3.02E-17 | Trans-ancestry |
| region_108 | rs144674978 | 15 | 33149751 | FMN1 | T/C | 0.013 | 1.34 (1.25–1.43) | 1.11E-18 | 1.23 (1.15–1.31) | 3.82E-10 | European |
| region_109 | rs3784710 | 15 | 68072458 | MAP2K5 | T/C | 0.763 | 1.05 (1.03–1.07) | 1.32E-07 | 1.05 (1.03–1.07) | 1.34E-08 | European |
| region_111 | rs12913420 | 15 | 90797010 | RP11-697E2.6 | C/G | 0.376 | 1.04 (1.03–1.06) | 2.29E-09 | NA | – | Trans-ancestry |
| region_114 | rs11117455 | 16 | 86179919 | RP11-805I24.4 | T/C | 0.181 | 1.04 (1.02–1.06) | 7.52E-06 | 1.05 (1.03–1.07) | 6.97E-07 | European |
| region_115 | rs73975588 | 17 | 816741 | NXN | A/C | 0.874 | 1.09 (1.07–1.12) | 6.62E-16 | 1.07 (1.04–1.09) | 6.08E-09 | European |
| region_117 | rs112592783 | 17 | 70633625 | LINCO0511 | T/C | 0.175 | 1.05 (1.03–1.07) | 5.87E-09 | 1.05 (1.03–1.07) | 5.95E-09 | Trans-ancestry |
| region_120 | rs4939821 | 18 | 46371993 | CTIF | T/C | 0.304 | 0.91 (0.89–0.92) | 4.12E-32 | 0.96 (0.94–0.97) | 1.89E-07 | European |

**Table 1 (continued) | Independent association signals uncovered at known CRC risk loci in conditional analyses (conditional P <1 × 10−6)**

| Fine-mapping region | SNP | Chr | Position | Nearby gene | Alleles | AF | Single-SNP analysis | | Joint analysis | | Group |
|---|---|---|---|---|---|---|---|---|---|---|---|
| | | | | | | | OR (95% CI) | P value[a] | OR (95% CI) | P value[b] | |
| region_121 | rs72971616 | 19 | 987366 | WDR18 | T/G | 0.063 | 0.92 (0.89–0.95) | 4.89E-06 | NA | – | European |
| region_126 | rs12460535 | 19 | 49098750 | SULT2B1 | A/G | 0.349 | 0.96 (0.95–0.98) | 5.76E-07 | NA | – | European |
| region_127 | rs8099852 | 19 | 58895221 | RPS5 | T/C | 0.546 | 1.03 (1.02–1.05) | 5.15E-07 | NA | – | Trans-ancestry |
| region_133 | rs1971480 | 20 | 48897080 | RP11-290F20.3 | T/G | 0.672 | 0.96 (0.95–0.98) | 8.61E-07 | 0.96 (0.94–0.97) | 6.50E-09 | European |
| region_133 | rs149942633 | 20 | 48983073 | RP11-290F20.2 | T/C | 0.153 | 1.12 (1.08–1.16) | 1.93E-08 | 1.1 (1.05–1.14) | 3.94E-06 | European |
| region_133 | rs6126008 | 20 | 49075315 | COX6CP2 | A/T | 0.660 | 0.96 (0.94–0.97) | 1.07E-08 | 0.96 (0.94–0.97) | 3.74E-09 | European |
| region_134 | rs34161672 | 20 | 56020599 | RBM38 | A/G | 0.321 | 1.04 (1.02–1.05) | 2.40E-06 | 1.04 (1.03–1.06) | 1.64E-07 | European |
| region_142 | rs78106213 | 22 | 46121230 | ATXN10 | T/G | 0.693 | 1.04 (1.03–1.06) | 2.41E-07 | 1.05 (1.03–1.06) | 8.20E-09 | European |

All independent association signals presented in this table are those not previously reported.
Chr and Position GRCh37, Alleles risk allele/Reference allele, AF Allele frequency, OR odds ratio, CI confidence interval.
[a]P value derived from trans-ancestry or ancestry-specific meta-analysis under the fixed-effects inverse variance weighted model.
[b]P value derived from conditional analysis conditioning on all other independent association signals in each fine-mapping region. "NA"—Only a single association signal was detected in the fine-mapping region in the analysis group.

STK39, IGFBP3, FUT2, and FUT3) showed nominal P < 0.05 significance considering only pLoF or combination of pLoF and Dmis, whereas no significance was detected for the remaining genes.

**Biological significance of the target genes for CCVs**
We utilized Enrichr[19–21] to analyze multiple pathway databases and identify enriched biological pathways among the 136 credible target genes (Methods). At a false-discovery rate (FDR) < 0.05, 126 pathways showed significant enrichment (Supplementary Data 22). Our findings were in line with our prior study[18] and highlighted the enriched signaling pathways such as TGF-β, BMP, Wnt, Hippo, and TNF-α/NF-κB, which are known to play a crucial role in the development and progression of colorectal cancer[19,20]. Of the 56 genes not previously reported, nine genes (TGIF1, CDKN2B, MYC, BMP7, WNT7B, PRICKLE2, LGR6, CEBPB, and IRS2) were mapped to these pathways (Table 5). Additionally, we identified several significant pathways, including those related to cancer, pluripotency of stem cells, epithelial–mesenchymal transition, extracellular matrix organization, adipogenesis, senescence, and autophagy in cancer. Interestingly, we also identified the glycolysis pathway, which provides energy support for cancer cells, as a significant pathway not previously reported. Four previously unreported genes, GOT1, IGFBP3, IRS2, and LCT, were mapped to glycolysis, supporting their association with CRC risk.

In addition, we performed functional annotation analysis on each credible target gene and assigned them to previously described cellular processes[18] (Supplementary Fig. 2). Of the 56 genes not previously reported, 26 were found to be involved in these cellular processes. Specifically, five genes were related to stemness/differentiation, one gene was linked to adhesion/migration, and six genes were associated with proliferation. Interestingly, we also identified an additional cellular process, post-translation modifications (PTMs) of protein, which included three genes (DACF12, USP12, and SENP8). These findings suggest potential critical roles of PTMs in the development of CRC.

## Discussion
Our study, including approximately 254,000 individuals of East Asian and European ancestry, represents the largest study conducted to fine-map CRC risk-associated genomic regions using GWAS data. We identified 238 independent association signals at conditional P value <1 × 10−6, including 47 signals not reported previously. Furthermore, integrating functional genomic data and results from cis-eQTL/mQTL and colocalization analyses, we identified 136 putative CRC susceptibility genes, including 56 genes that had not been previously reported. Notably, these identified genes are significantly enriched in several major CRC signaling pathways and other cancer-related pathways. Our findings not only significantly expanded the number of associated signals for CRC, but also provide substantial data to advance our understanding of CRC biology.

The integration of comprehensive functional genomic data from relevant colon tissues and cell lines, as well as genetic associations data, can facilitate the identification of potential target genes for CRC risk. Our study significantly extends previous efforts[7,11,16,17] by identifying 56 target gene candidates not previously reported for CRC risk, over half of which (29/56, 51.8%) are involved in the enriched biological pathways. For instance, eight target genes (TGIF1, CDKN2B, LGR6, MYC, PRICKLE2, WNT7B, BMP7, and TBX3) identified in this study may regulate normal intestinal homeostasis as they play roles in signaling pathways (i.e., Wnt and BMP) and pluripotency of stem cells. LGR6, for instance, is part of a G-protein-coupled receptor family and marks stem cells in the epidermis[22]. It activates a novel β-catenin/TCF7L2/LGR6-positive feedback loop in LGR6[high] cervical cancer stem cells (CSCs) to enhance the properties of cancer stem cells, including self-renewal, differentiation, and tumorigenicity[23]. Silencing of LGR6 resulted in the inhibition of stemness by repressing Wnt/β-catenin signaling in ovarian cancer[24]. TBX3, a transcriptional repressor, regulates stem cell maintenance by

 5

**Table 2 | Independent association signals with a single CCV**

| Fine-mapping region | SNP | Chr | Position | Alleles | AF | OR (95% CI) | P value[a] | Putative target gene(s)[b] |
|---|---|---|---|---|---|---|---|---|
| *European-specific analysis* | | | | | | | | |
| region_45 | rs116000952 | 6 | 32541270 | T/G | 0.843 | 0.92 (0.89–0.96) | 5.74E−06 | – |
| region_45 | rs6920820 | 6 | 30969938 | C/G | 0.980 | 0.84 (0.79–0.90) | 6.87E−08 | LINC00243 |
| region_61 | rs72681666 | 8 | 117641754 | T/C | 0.043 | 1.09 (1.05–1.13) | 1.57E−05 | – |
| region_62 | rs77569096 | 8 | 128468955 | A/G | 0.763 | 0.92 (0.90–0.94) | 2.06E−15 | – |
| region_84 | rs3217810 | 12 | 4388271 | T/C | 0.127 | 1.13 (1.11–1.16) | 1.96E−26 | – |
| region_108 | rs144674978 | 15 | 33149751 | T/C | 0.013 | 1.34 (1.25–1.43) | 1.11E−18 | – |
| region_133 | rs149942633 | 20 | 48983073 | T/C | 0.153 | 1.12 (1.08–1.16) | 1.93E−08 | – |
| *Trans-ancestry analysis* | | | | | | | | |
| region_1 | rs112191583 | 1 | 22554378 | T/C | 0.974 | 0.88 (0.83–0.92) | 1.19E−07 | – |
| region_24 | rs704417 | 3 | 64252424 | T/C | 0.546 | 1.05 (1.03–1.06) | 4.35E−10 | – |
| region_27 | rs113569514 | 3 | 133748789 | T/C | 0.763 | 1.08 (1.07–1.10) | 1.92E−21 | SLCO2A1 |
| region_29 | rs2578155 | 4 | 94836291 | C/G | 0.503 | 1.04 (1.03–1.06) | 1.09E−09 | – |
| region_42 | rs9379084 | 6 | 7231843 | A/G | 0.144 | 0.93 (0.91–0.95) | 2.39E−12 | RREB1 |
| region_46 | rs16878812 | 6 | 35569562 | A/G | 0.892 | 1.09 (1.07–1.12) | 7.62E−15 | FKBP5 |
| region_48 | rs6933790 | 6 | 41672769 | T/C | 0.788 | 1.08 (1.06–1.10) | 2.66E−20 | – |
| region_61 | rs4129064 | 8 | 117735666 | T/G | 0.734 | 1.06 (1.04–1.07) | 1.01E−09 | – |
| region_62 | rs6983267 | 8 | 128413305 | T/G | 0.508 | 0.86 (0.85–0.87) | 1.65E−122 | MYC |
| region_72 | rs704017 | 10 | 80819132 | A/G | 0.473 | 0.92 (0.91–0.93) | 1.97E−38 | – |
| region_84 | rs12818766 | 12 | 4376091 | A/G | 0.215 | 1.10 (1.08–1.12) | 1.81E−29 | – |
| region_89 | rs7398375 | 12 | 57540848 | C/G | 0.651 | 1.07 (1.05–1.09) | 3.70E−19 | LRP1 |
| region_94 | rs11067228 | 12 | 115094260 | A/G | 0.560 | 0.95 (0.94–0.97) | 2.50E−13 | – |
| region_96 | rs116964464 | 13 | 27543193 | T/C | 0.035 | 1.11 (1.07–1.15) | 4.83E−09 | USP12 |
| region_99 | rs7325844 | 13 | 73625133 | A/G | 0.639 | 1.05 (1.04–1.07) | 1.28E−12 | – |
| region_104 | rs35107139 | 14 | 54419106 | A/C | 0.550 | 0.92 (0.91–0.93) | 4.22E−36 | – |
| region_105 | rs8020436 | 14 | 59208437 | A/G | 0.370 | 1.06 (1.05–1.08) | 1.27E−17 | – |
| region_108 | rs17816465 | 15 | 33156386 | A/G | 0.193 | 1.09 (1.07–1.10) | 5.73E−20 | – |
| region_116 | rs1078643 | 17 | 10707241 | A/G | 0.765 | 1.09 (1.07–1.11) | 2.31E−27 | – |
| region_132 | rs6066825 | 20 | 47340117 | A/G | 0.662 | 1.08 (1.07–1.10) | 2.13E−32 | – |
| region_136 | rs1741640 | 20 | 60932414 | T/C | 0.208 | 0.88 (0.86–0.89) | 8.15E−55 | LAMA5, CABLES2 |

*Chr and Position* GRCh37, *Alleles* risk allele/Reference allele, *AF* Allele frequency, *OR* odds ratio, *CI* confidence interval. [a]P value derived from trans-ancestry or European-ancestry meta-analysis under the fixed-effects inverse variance weighted model; [b]"-" – No target genes were prioritized for the variant in this study.

controlling stem cell self-renewal and differentiation, and reduced expression levels of *TBX3* are associated with reduced pluripotency of stem cells[25,26]. *MYC* and *WNT7B* are implicated in the signaling related to the self-renewal and differentiation of cancer stem cells[27]. Here, we linked *MYC* and *WNT7B* with credible causal variants of CRC risk associations through functional genomic interaction. Our findings also indicated the relevance of glycolysis to CRC risk associations, a metabolic pathway critical in early CRC tumorigenesis by supporting the energetic and biosynthetic demands of CRC cells[28,29]. It should be noted that future studies are needed to validate chromatin interactions between identified CCVs and their target genes in this study by employing chromatin conformation capture technology such as in situ Hi-C, Capture Hi-C (CHi-C), and HiChIP.

Additional evidence supports some of the candidate target genes identified in our study as possible CRC susceptibility genes. In our differential gene expression analysis among normal colon mucosa, adenoma, and adenocarcinoma using gene expression data from 135 normal colon mucosas, 218 colon adenomas, and 2760 colon adenocarcinomas, we observed that 26 genes showed significant differential expression between adenoma and normal colon tissues, while 31 genes showed significant differential expression between carcinoma and adenoma tissues (adjusted *P* < 0.05) (Supplementary Data 20). Interestingly, three stemness/differentiation-related genes, including *LRRC34, CEBPB*, and *TBX3*, showed significant changes in their expression levels in adenoma compared to normal colon

mucosa. Additionally, 34 (60.7%) of not previously identified genes have been implicated in cancer-related functions in in vitro or in vivo functional experimental studies in CRC or other cancer types (Supplementary Data 20). These results provide further evidence supporting the potential involvement of these genes in CRC progression. Despite the above supportive evidence, it remains necessary to evaluate the functions of identified putative CRC susceptibility genes through both in vitro and in vivo assays in future investigations.

The trans-ancestry and ancestry-specific fine-mapping analyses conducted in this study not only enabled the discovery of independent association signals that are shared across populations of European and East Asian ancestry, but also revealed ancestry-specific signals. The larger sample size of the European-ancestry study enabled us to identify a larger number of independent association signals than the study conducted on Asians. However, there are some ancestry-specific signals identified in this study, which is most likely due to differences in LD structures and allele frequency between these two populations. Indeed, we observed distinct differences in the allele frequency for most ancestry-specific signals, as shown in Supplementary Data 4 and 5. For instance, the lead variant of 24 European ancestry-specific signals (40%, 24/60) is not detected among East Asian-ancestry populations. On the other hand, fine-mapping analyses capitalizing on ancestry differences in LD structure can substantially reduce the credible set size compared to European-ancestry specific analysis. This highlights the value of multi-ancestry fine-mapping over

**Table 3 | The 56 CRC susceptibility gene candidates not previously reported**

| Fine-mapping region | Gene | Lead variant | Distal | Proximal | Coding | Colocalization (eQTL) | Colocalization (mQTL) |
|---|---|---|---|---|---|---|---|
| region_1 | CELA3B | rs11579545 | | | | + | |
| region_1 | HSPG2 | rs11579545 | + | | | + | + |
| region_5 | PTGER3 | rs2651244 | | | | + | |
| region_7 | TNFSF18 | rs10489274 | | | | | + |
| region_9 | LGR6 | rs12122827 | | | | | + |
| region_10 | CNTN2 | rs12078075 | | + | | | + |
| region_12 | FMN2 | rs2078095 | | | | + | |
| region_14 | PPP1R21 | rs77272589 | | + | | + | |
| region_16 | LCT | rs1446585 | | | | | + |
| region_21 | GOLGA4 | rs1800734 | | | | + | |
| region_21 | MLH1 | rs1800734 | | + | | + | |
| region_24 | ADAMTS9 | rs6445418 | | | | | + |
| region_24 | PRICKLE2 | rs704417 | | | | + | |
| region_27 | SLCO2A1 | rs113569514 | | + | | | |
| region_28 | LRRC34 | rs10936599 | | | + | | |
| region_28 | ACTRT3 | rs10936599 | + | + | | | |
| region_28 | MYNN | rs10936599 | + | | + | | |
| region_34 | HPGD | rs1426947 | | | | + | |
| region_42 | LY86 | rs1294438 | | | | | + |
| region_44 | OR2I1P | rs73402748 | | | | + | |
| region_46 | SRPK1 | rs16878812 | | | | + | |
| region_49 | RUNX2 | rs57939401 | | | | | + |
| region_55 | IGFBP3 | rs80077929 | | | | | + |
| region_62 | MYC | rs4733655, rs6983267 | + | | | | |
| region_63 | CDKN2B | rs7859362 | + | | | | |
| region_63 | MTAP | rs7859362 | + | | | | |
| region_68 | VAV2 | rs7038489 | | | | | + |
| region_73 | KIF20B | rs140356782 | | | | + | |
| region_73 | PANK1 | rs140356782 | | | | + | + |
| region_74 | GOT1 | rs117746067 | | + | | | |
| region_75 | BORCS7 | rs12268849 | | | | + | |
| region_75 | AS3MT | rs12268849 | | + | | + | |
| region_79 | ANO1 | rs10751097 | | | | | + |
| region_92 | NTN4 | rs11108175 | | | | | + |
| region_93 | CUX2 | rs3858704 | | | | | + |
| region_94 | TBX3 | rs7300312, rs11067228 | + | | | | + |
| region_96 | USP12 | rs116964464 | + | | | | |
| region_101 | IRS2 | rs1078563 | | | | + | |
| region_101 | COL4A2 | rs4773184 | | | | | + |
| region_107 | BCL11B | rs80158569 | | | | + | |
| region_108 | GOLGA8N | rs56338436 | | | | + | |
| region_110 | SENP8 | rs8031386 | | + | | | + |
| region_111 | CIB1 | rs12913420 | | + | | + | |
| region_111 | ZNF774 | rs7179095 | + | | | | |
| region_119 | MYL12A | rs1612128 | + | | | | |
| region_119 | MYL12B | rs1612128 | + | | | | |
| region_119 | TGIF1 | rs1612128 | + | | | | |
| region_125 | B3GNT8 | rs1963413 | | | | + | |
| region_133 | CEBPB | rs1971480 | + | | | | |
| region_134 | RBM38 | rs34161672 | + | | | | |
| region_134 | BMP7 | rs6014965 | + | | | | + |
| region_138 | LSS | rs9983528 | | | | + | + |
| region_138 | PCNT | rs9983528 | | | + | + | |
| region_138 | SPATC1L | rs9983528 | | | | + | + |
| region_142 | WNT7B | rs62228060 | | | | | + |
| region_142 | ATXN10 | rs78106213 | | | | + | |

The lead variant for each gene is presented by independent association signals. Supporting evidence for the likely target gene is presented as follows: "Distal"—the CCV(s) located in distal enhancer elements of the gene; "Proxmial"—the CCV(s) located in proximal promoter element of the gene; "Coding"—the CCV is potential loss-of-function variants of the gene; "Colocalization (eQTL)"—target genes identified from eQTL colocalization analysis; "Colocalization (mQTL)"—target genes identified from mQTL colocalization analysis. "+" indicates the presense of supportive evidence.

**Table 4 | The 80 previously reported CRC susceptibility genes supported in this study**

| Fine-mapping region | Gene | Lead variant | Distal | Proximal | Coding | Colocalization (eQTL) | Colocalization (mQTL) |
|---|---|---|---|---|---|---|---|
| region_1 | WNT4 | rs6426749 | | | | + | |
| region_2 | FHL3 | rs61776719 | | | | + | + |
| region_8 | LAMC1 | rs8179460 | + | + | | | |
| region_9 | LMOD1 | rs12137232 | + | | | + | |
| region_15 | ACTR1B | rs11692435 | | | + | + | |
| region_18 | STK39 | rs4668039 | + | | | + | + |
| region_20 | TMBIM1 | rs3731861 | + | + | + | + | |
| region_23 | SFMBT1 | rs2001732, rs2581817 | | | | + | + |
| region_26 | BOC | rs73235124 | | + | | | |
| region_30 | TET2 | rs2047409, rs902443 | | | | + | + |
| region_31 | UGT8 | rs3924508 | | + | | | |
| region_35 | TERT | rs2735940 | | | | | + |
| region_40 | CDX1 | rs2302275 | | + | | | |
| region_41 | ERGIC1 | rs472959 | | | | | + |
| region_42 | RREB1 | rs9379084 | | | + | | |
| region_43 | EDN1 | rs2070699 | | | | | + |
| region_43 | HIVEP1 | rs4714081 | | | | + | + |
| region_47 | CDKN1A | rs9470361 | | | | | + |
| region_48 | TFEB | rs6933790 | | | | + | |
| region_52 | DCBLD1 | rs6911915 | | | | + | |
| region_53 | TCF21 | rs151127921 | + | | | | |
| region_54 | GNA12 | rs1182197 | | | + | + | + |
| region_55 | TBRG4 | rs67681615 | | | | | + |
| region_55 | TNS3 | rs6948177 | + | | | | |
| region_56 | ABHD11 | rs7806956 | | | | + | + |
| region_57 | TRIM4 | rs2527927 | | + | | + | |
| region_62 | POU5F1B | rs6983267 | | | | + | |
| region_64 | DCAF12 | rs11557154 | | | + | | |
| region_68 | BRD3 | rs11789898 | | | | + | + |
| region_70 | BAMBI | rs1773860 | + | | | | |
| region_71 | ASAH2B | rs10740013 | | | | + | |
| region_72 | ZMIZ1 | rs704017 | | | | | + |
| region_74 | ENTPD7 | rs35564340 | | | | + | |
| region_76 | TCF7L2 | rs4554812 | + | | | | |
| region_78 | TMEM258 | rs174570 | | + | | | |
| region_81 | TRPC6 | rs2186607 | | | | + | |
| region_82 | ARHGAP20 | rs3087967 | | | | + | |
| region_82 | FDX1 | rs3087967 | | | | + | |
| region_83 | BCL9L | rs497916 | + | | | | |
| region_85 | PLEKHG6 | rs10849434, rs1003563 | + | + | | | + |
| region_88 | CERS5 | rs11169572 | | | | | + |
| region_88 | ATF1 | rs11169572 | | + | | | + |
| region_88 | DIP2B | rs11169572 | | | | + | + |
| region_89 | LRP1 | rs7398375 | + | | | + | + |
| region_91 | TSPAN8 | rs11178634 | | | + | + | |
| region_98 | SMAD9 | rs12427846 | | + | | + | + |
| region_99 | KLF5 | rs1304959, rs78341008 | + | | | | |
| region_102 | NIN | rs1042266 | | | | + | |
| region_102 | ABHD12B | rs1042266 | | | | + | + |
| region_102 | PYGL | rs1042266 | | | | + | + |
| region_103 | NID2 | rs1151580 | | | | + | + |
| region_104 | BMP4 | rs1957628, rs35107139 | + | | | | + |
| region_105 | DACT1 | rs8020436 | | | | + | + |
| region_108 | GREM1 | rs16970016 | | | | | + |

**Table 4 (continued) | The 80 previously reported CRC susceptibility genes supported in this study**

| Fine-mapping region | Gene | Lead variant | Distal | Proxmial | Coding | Colocalization (eQTL) | Colocalization (mQTL) |
|---|---|---|---|---|---|---|---|
| region_109 | SMAD6 | rs3809570 | | + | | | + |
| region_109 | SMAD3 | rs56324967 | + | | | | |
| region_112 | ZFP90 | rs9924886 | | | | + | |
| region_112 | CDH1 | rs9924886 | + | + | | | + |
| region_115 | NXN | rs11247566 | | | | | + |
| region_117 | SOX9 | rs112592783 | + | | | | |
| region_118 | METRNL | rs35204860 | | | | + | + |
| region_120 | SMAD7 | rs4939821, rs2337113 | + | | | | + |
| region_122 | FUT3 | rs10409772 | | | + | + | |
| region_124 | RHPN2 | rs28840750 | + | | | + | |
| region_126 | FUT2 | rs12460535 | | | + | + | |
| region_127 | TRIM28 | rs11670192 | | | | + | |
| region_127 | ZNF584 | rs8099852, rs11670192 | | + | | + | |
| region_128 | BMP2 | rs990999 | | | | + | |
| region_130 | MAP1LC3A | rs6059938 | | | | + | |
| region_130 | MYH7B | rs6059938 | | | + | | |
| region_131 | TOX2 | rs6073241 | | | | + | + |
| region_132 | PREX1 | rs6066825 | | | | | + |
| region_133 | PARD6B | rs6091213 | | | | + | |
| region_133 | PTPN1 | rs6091213 | + | | | | |
| region_135 | GNAS | rs8121252 | | + | | | |
| region_136 | RBBP8NL | rs1741640 | | | | + | |
| region_137 | STMN3 | rs6089763 | | | | | + |
| region_139 | ZNRF3 | rs4616575 | + | | | + | |
| region_140 | PDGFB | rs130651 | | | | | + |
| region_142 | RIBC2 | rs6007600 | | | | + | + |

The lead variant for each gene is presented by independent association signals. Supporting evidence for the likely target gene is presented as follows: "Distal"—the CCV(s) located in distal enhancer elements of the gene; "Proxmial"—the CCV(s) located in proximal promoter element of the gene; "Coding"—the CCV is potential loss-of-function variants of the gene; "Colocalization (eQTL)"—target genes identified from eQTL colocalization analysis; "Colocalization (mQTL)"—target genes identified from mQTL colocalization analysis. "+" indicates the presense of supportive evidence.

single-ancestry analysis. Our analysis is limited to two ancestry groups. Further studies should increase the diversity of genetic data, including those from other racial groups.

In summary, our large trans-ancestry fine-mapping analysis has identified large numbers of not previously reported independent association signals for CRC risk and refined the majority of the previously reported association signals. By leveraging data from two ancestries, we further defined putative causal variants underlying CRC risk signals. Our study has also uncovered a credible set of target genes. These findings offer a significant advancement in our understanding of the genetic and biological processes underlying CRC and provide a roadmap for further investigation of variants and genes identified in our study.

## Methods
### GWAS data and meta-analysis
The GWAS data used in this study comprised 100,204 CRC cases and 154,587 controls (Supplementary Data 1), which were grouped into 31 GWAS analytical units based on the study or genotyping platform as consistent with the original reports. Of them, 17 datasets were derived from populations of European descent and 14 were from populations of Asian descent. These 31 GWAS datasets were meta-analyzed under the fixed-effects inverse variance weighted model implemented in METAL[30]. Further details regarding each analytical unit and meta-analysis were described in Supplementary Note.

### Identifying independent association signals
A total of 205 independent genetic associations have been reported for CRC risk by GWAS[7]. To define fine-mapping regions for CRC, we aggregated these risk variants using *bedtools*. Specifically, we identified 1 megabase (Mb) intervals centered on the risk variants, and if there were regions of overlap, we combined them into a single interval over 1 Mb. In total, we determined 143 fine-mapping regions, including 142 on autosomes and one on chromosome X (Supplementary Data 2). Our fine-mapping analysis and downstream analyses focused on the 142 genomic risk regions on autosomes.

To identify distinct association signals within each risk region, we conducted a forward stepwise conditional analysis for summary statistics from the trans-ancestral meta-analysis, using GCTA-COJO[31,32]. We included common variants (MAF > 0.01) with associations at $P < 0.05$ in both populations. To account for differences in the LD structure, we conducted conditional analysis in each population for each fine-mapping region, conditioning on the most significant association from the trans-ancestral summary statistics. We then meta-analyzed the conditioned results using the fixed-effects inverse variance weighted model with METAL. To identify potential ancestry-specific independent signals, we also performed conditional analysis in each population, conditioning on the ancestry-specific most significant association. Common variants (MAF > 0.01) with association at $P < 1 \times 10^{-4}$ in each population were included. For LD estimation, we used genotyping data from 6684 unrelated samples of Asian descent[33], and 503 European samples in the 1000 Genome project as the reference.

Following a previous study conducted for breast cancer[12], we applied the conditional $P$ value $< 1 \times 10^{-6}$ to define the independent signal. For each region, we first adjusted for the most significant association and then added any additional variant that remained an independent signal at the conditional $P$ value $< 1 \times 10^{-6}$ to the

**Table 5 | Significant enrichment in biological pathways**

| Pathways[a] | Genes[b] |
|---|---|
| TGF-beta signaling | *BAMBI, BMP2, BMP4,* **BMP7,** *CDH1,* **CDKN2B,** *GREM1,* **MYC, RUNX2,** *SMAD3, SMAD6, SMAD7, SMAD9,* **TGIF1** |
| Hippo signaling | *BMP2, BMP4,* **BMP7,** *CDH1, GNAS,* **MYC,** *PARD6B, SMAD3, SMAD7, TCF7L2, WNT4,* **WNT7B** |
| TNF-alpha Signaling via NF-kB | **TGIF1,** *BMP2, CDKN1A, EDN1,* **CEBPB,** *SMAD3,* **MYC, IRS2** |
| BMP signaling | *BMP2, SMAD6,* **RUNX2,** *SMAD9, SMAD7* |
| Pluripotency of stem cells | *POU5F1B, BMP4, SMAD3,* **MYC, WNT7B,** *SMAD9,* **TBX3,** *WNT4, PDGFB, SMAD6, SMAD7, TCF7L2* |
| Epithelial–mesenchymal transition | *SMAD3, CDH1,* **RUNX2,** *GREM1,* **COL4A2,** *LRP1,* **IGFBP3,** *LAMC1, NID2,* **WNT7B,** *WNT4* |
| Extracellular matrix organization | *BMP4, BMP2,* **COL4A2,** *PDGFB,* **NTN4,** *LAMC1,* **HSPG2,** *NID2,* **BMP7, ADAMTS9** |
| Senescence and Autophagy | *BMP2, CDKN1A,* **CEBPB,** *SMAD3, MAP1LC3A,* **IGFBP3, CDKN2B, MYC,** *KLF5* |
| DNA damage response | *TCF7L2, CDKN1A, SMAD3,* **MYC, WNT7B,** *WNT4* |
| Cell cycle | *CDKN1A,* **CDKN2B,** *SMAD3,* **MYC** |
| Focal adhesion | **COL4A2,** *PDGFB, LAMC1,* **MYL12A, MYL12B, VAV2** |
| Adherens junction | *PTPN1, TCF7L2, SMAD3, CDH1* |
| Glycolysis | **GOT1, IGFBP3, IRS2,** *SOX9, PYGL,* **LCT** |
| Proteoglycans in cancer | *CDKN1A,* **MYC, WNT7B, HSPG2,** *WNT4,* **VAV2** |
| Androgen Response | **HPGD,** *ZMIZ1, STK39,* **MYL12A** |
| Sphingolipid Metabolism | *UGT8, CERS5* |
| Other cancer related pathways[c] | *TCF7L2, CDKN1A, EDN1,* **CDKN2B,** *SMAD3,* **WNT7B,** *PTGER3, PDGFB, LAMC1,* **MLH1,** *BMP4, BMP2,* **COL4A2,** *TERT, CDH1,* **MYC,** *GNA12, GNAS, WNT4, ATF1,* **CEBPB, BCL11B, HPGD, IGFBP3, RUNX2,** *ZMIZ1, SMAD6, SMAD7,* **VAV2,** *TFEB* |

[a]Genes from the same or similar pathway item in multiple databases were combined.
[b]Genes identified in this study for each pathway item are highlighted in bold.
[c]Genes from all general cancer-related pathways (i.e., pathway in cancer, colorectal cancer) identified in multiple databases were combined.

conditional set. We then repeated the conditional analysis until no more variants met the significance threshold. In regions with multiple independent signals, we determined the index variant for each signal through a process of conditional analysis, adjusting for the index variants of the other signals. This process was repeated until the set of index variants were stabilized. The variant with the strongest residual association was defined as the index for the signal.

For independent association signals identified in ancestry-specific analyses, we compared them with those from trans-ancestry analyses by assessing correlations between their lead variants within each risk region. If a signal was consistently found in both ancestry-specific and trans-ancestry analyses (i.e., the same lead variant or correlated lead variants with LD $r^2 > 0.1$ in each corresponding population), we considered it as a sharing signal between Asian and European-ancestry populations. Otherwise, they were defined as ancestry-specific signals.

### Identifying a set of CCVs of each independent signal

To determine the CCVs of each independent signal, we used the approach described in a previous study for breast cancer[12]. Specifically, variants that have a conditional *P* value within two orders of magnitude of the most significant association, conditioning on all other independent association signals, were defined as CCVs.

### RNA-seq data analysis

We conducted mRNA sequencing on tumor-adjacent normal colon tissues obtained from 364 East Asians patients with colorectal cancer who participated in the ACCC. Furthermore, we included RNA-seq data from normal colon tissues from 423 individuals of European ancestry who participated in the BarcUVa-Seq project. Included subjects, library preparation and sequencing of colon tissue samples in the ACCC and the BarcUVa-Seq project have been presented in Supplementary Note.

The raw RNA-seq data were processed according to the pipeline of the GTEx Consortium. Sequencing reads were aligned to the reference genome GRCh37 (RNA-seq data from East Asians) or GRCh38 (RNA-seq data from the BarcUVa-Seq project) with STAR (v2.5.4)[34]. Quality control of aligned samples was performed using RNA-SeQC (v2.3.5)[35]. Samples that met any of the following criteria were removed: (1) <10 million mapped reads; (2) read mapping rate < 0.2; (3) intergenic mapping rate

>0.4; (4) base mismatch rate >0.01 for read mate 1 or >0.02 for read mate 2; and (5) rRNA mapping rate >0.3. If the sample had replicated RNA-seq data, the one with the highest mapped reads was retained.

Gene-level expression quantification was performed using RNA-SeQC based on the GENCODE release 19 annotation (for RNA-seq data from East Asians) and the GENCODE release 26 annotation (for RNA-seq data from the BarcUVa-Seq project)[36]. The read counts and TPM values of genes were calculated using aligned reads with the following criteria: (1) reads were uniquely mapped; (2) aligned reads were properly paired; (3) the read alignment distance was <6. The genes with expression thresholds of ≥0.1 TPM in ≥20% of samples and ≥6 reads (unnormalized) in ≥20% of samples were selected. Quantile normalization of the gene expression was performed. We further performed rank-based inverse normal transformation for the expressions of genes across samples.

### Cis-expression/methylation quantitative loci (cis-eQTL/mQTL) analysis

To identify target genes, we performed cis-eQTL analysis based on a linear regression framework[10,11]. Gene expression data from four expression datasets comprising a total of 1,299 individuals were used: 1) GTEx project of transverse colon tissues from 368 individuals predominantly of European ancestry, 2) Colonomics project of normal colon tissues or tumor-adjacent normal colon tissues from 144 individuals of European ancestry, 3) BarcUVa-Seq project of normal colon tissues from 423 individuals of European ancestry, and 4) ACCC of tumor-adjacent normal colon tissue from 364 CRC patients of East Asian ancestry. We obtained available cis-eQTL results for CCVs and their nearby genes (within 1 Mb to CCV) from the GTEx database (version 8) and the Colonomics project. Details for gene expression data and eQTL analysis in the Colonomics project are explained elsewhere[37]. For the analyses using the remaining two datasets, we conducted a linear regression analysis to assess the associations between CCV and the normalized expression levels of nearby genes (within 1 Mb to CCV), adjusting for age, gender, and five top principal components.

We conducted cis-mQTL analysis for CCVs identified in European and trans-ancestry analyses. To do this, we included methylation data

obtained from a total of 321 individuals. These datasets consisted of 189 transverse colon tissues predominantly of European ancestry from GTEx, as well as normal colon tissues or tumor-adjacent normal colon tissues of 132 individuals of European ancestry from the Colonomics project. We extracted cis-mQTL results for CCVs and their nearby CpG sites (within 1 Mb to CCV) from the GTEx database (version 8)[14]. In the Colonomics project, a linear regression analysis was used to evaluate the associations between CCV and the normalized methylation levels of CpG sites (within 1 Mb to CCV), with adjustments of age, gender, and colon sites (right/left). Further details about the cis-mQTL in the Colonomics project can be found in previous studies[37,38].

## Meta-analysis of cis-eQTL/mQTL results

We performed a meta-analysis to integrate the summary cis-eQTL/mQTL results based on beta and p values from different datasets[10,11]. In brief, we calculated the z score from function qnorm(p/2)*sign(beta) and further converted the standard z score derived from sum(z*sqrt(N))/sqrt(sum(N)) with a normalized weighted sampled size. Here, beta and $p$ value were derived from eQTL/mQTL results and N referred to the sample size for each dataset. The meta p value was derived from the standard z score. For independent signals detected in both European and Asian populations, the eQTL results from both populations were combined.

We adjusted the combined p-values of eQTL/mQTL results with the Bonferroni procedure. The procedure was conducted for index variants of independent association signals. The Bonferroni-adjusted $P < 0.05$ was applied to identify potential target genes for each signal.

## Colocalization analyses between GWAS association signals and eQTL/mQTL signals

To identify putative target genes, we employed the SMR method to conduct a colocalization analysis[39]. We integrated GWAS summary statistics of CCVs and their associations with genes from eQTL/mQTL analysis described above. The results of meta-analyses on cis-eQTLs/mQTLs were used. Specifically, we have a statistic:

$$T_{SMR} = b_{xy}^2 / Var\left(b_{xy}\right) \approx \frac{Z_{zy}^2 Z_{zx}^2}{Z_{zy}^2 + Z_{zx}^2} \qquad (1)$$

Here, $Z_{zx}$ and $Z_{zy}$ are the Z statistics for the GWAS summary statistics and the cis-eQTL/mQTL results, respectively. $T_{SMR}$ is the χ2 statistic, which tests the significance of $b_{xy}$. The significant colocalized signals were determined based on the threshold of the Bonferroni-corrected $P_{SMR} < 0.05$ within each independent signal.

## Functional annotation of CCVs

We investigated whether each potential causal variant was mapped to gene regulatory regions (i.e., promoter or enhancer) (Supplementary Data 8). We obtained 351 chromatin immunoprecipitation sequencing (ChIP-seq) peak files for histone modification marks and transcription factors, and 25 DNase I hypersensitive sites sequencing peak files for chromatin accessibility, generated in normal colorectal epithelium and CRC cell lines from the Cistrome database[40,41]. Only peaks that met all six quality controls set recommended by Cistrome were analyzed. Additionally, we obtained available ChIP-seq data of histone modification marks from colon tissues, tumor tissues of CRC, and CRC cell lines from Gene Expression Omnibus (GEO), which included 16 from GSE133928[42], 215 from GSE136889[43] and 233 from GSE156613[44]. To generate coverage tracks Bigwig (bw) files for ChIP-seq data, we converted them to bedGraph files and then identified peaks with the subcommand *bdgpeakcall* from macs2[45]. For each variant, we examined whether it was mapped to a peak region of histone modification marks, DNase I hypersensitive, or transcription factors binding sites using an *in-house* script.

## In silico prediction of regulatory element-to-gene

Since the majority of the CCVs are located outside protein-coding regions, genes can potentially be regulated by CCVs located in distal enhancer elements and proximal promoter elements. Hence, we identified an extensive set of functional genomic data from normal colon tissues or tumor tissues of colorectal cancer or colorectal cancer cell lines (Supplementary Data 9). Subsequently, we conducted an in-silico analysis for each CCV-gene pair.

We used a variety of experimental and computational functional genomic data to identify target genes of CCVs in regulatory elements. Specifically, for distal regulatory elements, we utilized chromatin-chromatin interaction data from experiments or computational predictions. To do this, we downloaded 13 experimental chromatin-chromatin interaction datasets under accessions GSE133928[42] and GSE136629[43] from GEO, as well as two promoter capture Hi-C datasets from the previous study[46]. We combined this data with ChIP-seq data of the histone modification H3K27ac (an active enhancer mark) to identify enhancer-promoter loops. We defined these loops as interactions where one fragment overlapped an H3K27ac peak (enhancer-like) and the other fragment overlapped the promoter of a gene (the region from downstream 1 kb to upstream 100 bp around the transcription start site).

In addition to this, we downloaded experimentally confirmed enhancer-gene pairs from the ENdb database. We also obtained computational enhancer-promoter interactions from IM-PET[47], FANTOM5[48,49], EnhancerAtlas[50], and super-enhancer[51,52]. To further refine our analysis, we included topologically associating domain (TAD) boundaries in three colorectal cancer cell lines (HT29, LoVo, and DLD1)[46,53]. Finally, we examined the overlap between CCVs and enhancer elements. For proximal promoter elements, we analyzed CCVs located within gene promoter regions that intersected with ChIP-seq peaks of H3K4me3 (an activity promoter mark).

To identify potential loss-of-function variants and their corresponding targeted genes, we conducted variant annotation of CCVs using the Variant Effect Predictor (VEP) tool[54]. To predict the consequence of missense coding variants, we utilized PolyPhen-2 and SIFT. Furthermore, to evaluate splicing effects, MaxEntScan was used.

We scored CCV target genes using different criteria (Supplementary Data 9). For the potential target gene of CCV in distal enhancer elements, the gene was awarded two points or one point if there was evidence from experimental chromatin-chromatin interaction or computed interaction. The score was unweighted to three if both experimental and computational interaction were detected for the gene-CCV pair. If CCV interacted with genomic features (open chromatin, activity enhancer, and TF binding sites), the corresponding gene was further unweighted by one point. An additional point was awarded if there are at least two interactions for the CCV. If the gene were colorectal cancer or pan-cancer drivers[55], they were up-weighted by an additional point. The score was down-weighted for the gene if the CCV-gene pair was separated by TAD or a lack of expression in colon tissues. Distal scores eventually ranged from 0 to 6. For the potential target gene of CCV in proximal promoter elements, the gene was awarded one point if CCV overlapped with binding sites of transcription factors. If genes were colorectal cancer or pan-cancer drivers, they were up-weighted with an additional point. A lack of its expression resulted in down-weighting to 10% as target genes. Proximal scores eventually ranged from 0 to 2. Genes predicted to be regulated targets of coding CCVs were awarded points based on the annotation as either of missense, nonsense, and predicted splicing alterations. The consequences of missense variants which probably are damaging or deleterious resulted in the addition of one point to the target gene. Further points were awarded to such a gene if it was colorectal cancer or pan-cancer drivers. A lack of expression reduced the score (the score was down to 10%). Coding scores ranged from 0 to 2. For the set of confident target genes, we defined such genes if it has a distal score >4 or a proximal score >1, or a coding score >1.

## Credible set of susceptibility genes

To determine a set of credible genes for CRC susceptibility, we combined information on gene-CRC risk associations through TWAS and colocalization of eQTL signal with GWAS risk signals for genes that were present in both our study and previous investigations. We used three sets of previously identified genes below: (A) 155 effector genes identified through GWAS, TWAS, TIsWAS, and MWAS[7]; (B) 136, 26, and 48 genes identified through TWASs[7,16,17]; (C) 73 genes identified through colocalization analysis between eQTL and GWAS signals[11] or genes associated with CRC risk at nominal $P < 0.05$ in the previous TWAS[17]. We considered the prioritization order as A > B > C for these three gene sets and focused on protein-coding genes outside the MHC region. For the independent association signals with multiple target gene candidates, we kept either genes with higher prioritization or all genes if there was no evidence from these three gene sets. For the independent association signals with a single gene, we kept it regardless of evidence from the gene sets.

## Single-cell RNA-sequencing data analysis

We included single-cell RNA-sequencing datasets from colon tissues of 31 individuals who participated in the Colorectal Molecular Atlas Project (COLON MAP)[18]. We analyzed gene expression dataset for each individual's cell and combined these datasets into a count matrix. We normalized the number of unique molecular identifiers (UMIs) per cell and converted it to transcripts per 10,000 transcripts (TP10K). Next, we applied a logarithmic transformation to the normalized values and got the $\log_2(\text{TP10K} + 1)$ expression matrix for the downstream analyses. Further, we determined the 2000 most highly variable genes within the entire dataset and performed a principal component analysis (PCA). The top 30 and 40 principal components (PCs) were identified. Subsequently, we performed batch correction removal and utilized the top 40 batch-corrected components to construct a k-nearest neighbors graph of cell profiles with $k = 9$. We visualized the individual single-cell profiles using the Uniform Manifold Approximation and Projection (UMAP) and constructed the neighborhood graph using the Leiden graph-clustering method. Nine cell types were defined, including well-known major cell types such as absorptive cells (ABS), crypt top colonocytes (CT), enteroendocrine cells (EE), goblet cells (GOB), stem cells (STM), and others. We identified differentially expressed genes (DEGs) by comparing each cell type with all other cell types and calculated a $P$-value for each gene using Wilcoxon's rank-sum test. The criteria |log2 fold change (FC)| > 1 and $P < 0.05$ were applied to determine genes with significantly differential expression between cell types.

## Burden test for credible susceptibility genes

We annotated all variants in the UKBB WES 200 K cohort with functional annotations from ANNOVAR[56] based on the reference genome GRCh38. We only included rare loss-of-function (LoF) and deleterious missense (Dmis) variants with MAF < 0.01 in our gene-based test. LoF variants were those predicted as frameshift insertion/deletion, splice-site alteration, stop gain, and stop loss by ANNOVAR, and deleterious missense (Dmis) variants were those predicted as deleterious by MetaSVM[57]. We considered both LoF sets and damaging sets (LoF+ Dmis) within a gene for testing. For a given set, we collapsed rare variants within a gene as a single combined 'mask' and tested the association between the 'mask' genotype and the CRC phenotype using logistic regression after adjusting for sex, age, the interaction of sex and age, and the top four principal components.

## Pathway analysis of credible susceptibility genes

To explore the potential biological roles of the identified CRC susceptibility genes, we analyzed their functional enrichment using the enrichR[19–21] and various pathway databases, including WikiPathway, KEGG, MSigDB, and Reactome. The biological pathways (adjusted $P < 0.05$) were considered and presented.

## Reporting summary

Further information on research design is available in the Nature Portfolio Reporting Summary linked to this article.

## Data availability

The GWAS summary statistics are available at the GWAS catalog under accession number GCST90129505. The RNA-seq data and genotype data of subjects of East Asian ancestry from the ACCC is being deposited to NCBI database of Genotypes and Phenotypes (dbGaP, accession number phs002813.v1.p1). All requests to access these data could also be made by contacting Drs. Wei Zheng (wei.zheng@vanderbilt.edu) and Xingyi Guo (xingyi.guo@vumc.org). The data from the Genotype-Tissue Expression (GTEx, version 8) project used in this study are publicly available at the dbGaP under accession number phs000424.v8.p2 (https://www.ncbi.nlm.nih.gov/projects/gap/cgi-bin/study.cgi?study_id=phs000424.v8.p2). The transcriptome and genotype data as well as the sample covariates from the BarcUVa-Seq project can be accessed at the dbGaP under accession number phs003338.v1.p1 (https://www.ncbi.nlm.nih.gov/projects/gap/cgi-bin/study.cgi?study_id=phs003338.v1.p1). The access to data from the Colonomics project could be requested by submission of an inquiry to Dr. Victor Moreno (v.moreno@iconcologia.net). The CRC-relevant epigenome and functional genomic data were obtained from the NCBI's Gene Expression Omnibus database (GEO) under accession numbers: GSE133928, GSE136889, and GSE156613. Enhancer-promoter interaction data were obtained from the ENdb database (https://bio.liclab.net/ENdb/), 4Dgenome (https://4dgenome.research.chop.edu/), FANTOM5 (https://fantom.gsc.riken.jp/5/), EnhancerAtlas 2.0 (http://www.enhanceratlas.org/) and Super-enhancers (https://bio.liclab.net/sedb/ and https://www.cell.com/fulltext/S0092-8674(13)01227-0#supplementaryMaterial). Single-cell RNA-sequencing datasets from colon tissues of 31 individuals were obtained from the Colorectal Molecular Atlas Project (COLON MAP). Whole exome sequencing data from 137,104 individuals of European ancestry were obtained from the UK Biobank (https://www.ukbiobank.ac.uk/).

## Code availability

The code used in this study is available at the GitHub repository https://github.com/zhishanchen/CRC_Finemapping[58].

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

## Acknowledgements

This research was supported primarily by US National Institutes of Health (NIH) grant R01CA188214 (to W.Z.), Anne Potter Wilson Chair endowment from the Vanderbilt University School of Medicine (to W.Z.), and NIH grant R37CA227130 and R01CA269589 (to X.G.). Sample preparation and genotyping assays at Vanderbilt University were conducted at the Survey and Biospecimen Shared Resources and Vanderbilt Microarray Shared Resource, supported in part by the Vanderbilt-Ingram Cancer Center (grant P30CA068485). Data analyses were performed on servers maintained by the Advanced Computing Center for Research and Education (ACCRE) at Vanderbilt University.

## Author contributions

Wei Zheng and Xingyi Guo conceived and supervised the study, and acquired funding. Wei Zheng, Xingyi Guo, and Zhishan Chen designed the study with significant contribution from Ran Tao. Zhishan Chen carried out the main analysis. Chao Li, Quanhu Shen, and Ken S Lau contributed to single-cell RNA-seq analysis. Yuhan Xie and Hongyu Zhao contributed to whole exome sequencing analysis. Zhishan Chen, Xingyi Guo, and Wei Zheng interpreted results with help from other authors. Zhishan Chen, Xingyi Guo, Jeroen R Huyghe, Philip J Law, Ceres Fernandez-Rozadilla, Jie Ping, Guochong Jia, Maria N Timofeeva, Minta Thomas, Stephanie L Schmit, Virginia Díez-Obrero, Matthew Devall, Ferran Moratalla-Navarro, Juan Fernandez-Tajes, Sarah E W Briggs, Victoria Svinti, Kevin Donnelly, Yingchang Lu, Fredrick R Schumacher, Stephanie J Weinstein, Kala Visvanathan, Kostas K Tsilidis, Yu-Ru Su, Robert Steinfelder, Sonja I Berndt, Sushma S Thomas, Kimberly F Doheny, Tameka Shelford, Amit D Joshi, Anshul Kundaje, Christopher K Edlund, Andre Kim, Lori C Sakoda, Stephanie A Bien, Yi Lin, Conghui Qu, Chenxu Qu, Stuart Reid, and Li Hsu analyzed the data. Xingyi Guo, Ceres Fernandez-Rozadilla, Jirong Long, Matthew Devall, Claire Palles, Kitty Sherwood, Susan M Farrington, James Blackmur, Peter G. Vaughan-Shaw, Xiao-Ou Shu, Peter Broderick, James Studd, Tabitha A Harrison, David V Conti, Marilena Melas, Gad Rennert, Mireia Obón-Santacana, Vicente Martín-Sánchez, Jae Hwan Oh, Jeongseon Kim, Sun Ha Jee, Keum Ji Jung, Sun-Seog Kweon, Min-Ho Shin, Aesun Shin, Yoon-Ok Ahn, Dong-Hyun Kim, Isao Oze, Wanqing Wen, Keitaro Matsuo, Koichi Matsuda, Chizu Tanikawa, Zefang Ren, Yu-Tang Gao, Wei-Hua Jia, John L Hopper, Mark A Jenkins, Aung Ko Win, Rish K Pai, Jane C Figueiredo, Robert W Haile, Steven Gallinger, Michael O Woods, Polly A Newcomb, David Duggan, Jeremy P. Cheadle, Richard Kaplan, Rachel Kerr, David Kerr, Iva Kirac, Jan Böhm, Jukka-Pekka Mecklin, Pekka Jousilahti, Paul Knekt, Lauri A. Aaltonen, Harri Rissanen, Eero Pukkala, Johan G Eriksson, Tatiana Cajuso, Ulrika Hänninen, Johanna Kondelin, Kimmo Palin, Tomas Tanskanen, Laura Renkonen-Sinisalo, Satu Männistö, Demetrius Albanes, Edward Ruiz-Narvaez, Julie R Palmer, Daniel D Buchanan, Elizabeth A Platz, Cornelia M Ulrich, Erin Siegel, Stefanie Brezina, Andrea Gsur, Peter T Campbell, Jenny Chang-Claude, Michael Hoffmeister, Hermann Brenner, Martha L Slattery, John D Potter, Matthias B Schulze, Marc J Gunter, Neil Murphy, Antoni Castells, Sergi Castellví-Bel, Leticia Moreira, Volker Arndt, Anna Shcherbina, D. Timothy Bishop, Graham G Giles, Melissa C. Southey, Gregory E Idos, Kevin J McDonnell, Zomoroda Abu-Ful, Joel K Greenson, Katerina Shulman, Flavio Lejbkowicz, Kenneth Offit, Temitope O Keku, Bethany van Guelpen, Thomas J Hudson, Heather Hampel, Rachel Pearlman, Richard B Hayes, Marie Elena Martinez, Paul D. P. Pharoah, Susanna C Larsson, Yun Yen, Heinz-Josef Lenz, Emily White, Li Li, Elizabeth Pugh, Andrew T Chan, Marcia Cruz-Correa, Annika Lindblom, David J Hunter, Clemens Schafmayer, Peter C Scacheri, Robert E Schoen, Jochen Hampe, Zsofia K Stadler, Pavel Vodicka, Ludmila Vodickova, Veronika Vymetalkova, W. James Gauderman, David Shibata, Amanda Toland, Sanford Markowitz, Stephen J Chanock, Franzel van Duijnhoven, Edith JM Feskens, Manuela Gago-Dominguez, Alicja Wolk, Barbara Pardini, Liesel M FitzGerald, Soo Chin Lee, Shuji Ogino, Charles Kooperberg, Christopher I Li, Ross Prentice, Stéphane Bézieau, Taiki Yamaji, Norie Sawada, Motoki Iwasaki, Loic Le Marchand, Anna H Wu, Caroline E McNeil, Gerhard Coetzee, Caroline Hayward, Ian J Deary, Sarah E Harris, Evropi Theodoratou, Marion Walker, Li Yin Ooi, Qiuyin Cai, Malcolm G Dunlop, Stephen B Gruber, Richard S Houlston, Victor Moreno, Graham Casey, Ulrike Peters, Ian Tomlinson, and Wei Zheng recruited patients and collected samples. Zhishan Chen, Xingyi Guo, and Wei Zheng wrote the manuscript with substantial contributions from Ceres Fernandez-Rozadilla, Jie Ping, Guochong Jia, Jirong Long, Xiao-Ou Shu, Richard S Houlston, and Ian Tomlinson. All authors have reviewed and approved the final manuscript.

## Competing interests

Antoni Castells is a consultant to Bayer Pharma AG, Boehringer Ingelheim and Pfizer Inc. for work unrelated to this manuscript. Anna Shcherbina is an employee at Insitro, including consulting fees from BMS. Heather Hampel is SAB for Invitae Genetics, Promega and Genome Medical, Stock/Stock options for Genome Medical and GI OnDemand. Rish K Pai collaborates with Eli Lilly, AbbVie, Allergan, Verily and Alimentiv, which includes consulting fees (outside the submitted work). Stephanie A Bien has a financial interest in Adaptive Biotechnologies. Stephen B Gruber is co-founder, Brogent International LLC. One of Zsofia K Stadler's immediate family members serves as a consultant in ophthalmology for Alcon, Adverum, Gyroscope Therapeutics Limited, Neurogene and RegenexBio (outside the submitted work). Victor Moreno has research projects and owns stocks of Aniling. The remaining authors declare no competing interests.

## Additional information

Zhishan Chen[1], Xingyi Guo[1,2], Ran Tao[3,4], Jeroen R. Huyghe[5], Philip J. Law[6], Ceres Fernandez-Rozadilla[7,8], Jie Ping[1], Guochong Jia[1], Jirong Long[1], Chao Li[1], Quanhu Shen[3], Yuhan Xie[9], Maria N. Timofeeva[10,11], Minta Thomas[5], Stephanie L. Schmit[12,13], Virginia Díez-Obrero[14,15,16,17], Matthew Devall[18], Ferran Moratalla-Navarro[14,15,16,17], Juan Fernandez-Tajes[7], Claire Palles[19], Kitty Sherwood[7], Sarah E. W. Briggs[20], Victoria Svinti[10], Kevin Donnelly[10], Susan M. Farrington[10], James Blackmur[10], Peter G. Vaughan-Shaw[10], Xiao-Ou Shu[1], Yingchang Lu[1], Peter Broderick[6], James Studd[6], Tabitha A. Harrison[5], David V. Conti[21], Fredrick R. Schumacher[22,23], Marilena Melas[24], Gad Rennert[25,26,27], Mireia Obón-Santacana[14,15,17], Vicente Martín-Sánchez[15,28], Jae Hwan Oh[29], Jeongseon Kim[30], Sun Ha Jee[31], Keum Ji Jung[31], Sun-Seog Kweon[32], Min-Ho Shin[32], Aesun Shin[33,34], Yoon-Ok Ahn[34], Dong-Hyun Kim[35], Isao Oze[36], Wanqing Wen[1], Keitaro Matsuo[37,38], Koichi Matsuda[39], Chizu Tanikawa[40], Zefang Ren[41], Yu-Tang Gao[42], Wei-Hua Jia[43], John L. Hopper[44,45], Mark A. Jenkins[44], Aung Ko Win[44], Rish K. Pai[46], Jane C. Figueiredo[21,47], Robert W. Haile[48], Steven Gallinger[49], Michael O. Woods[50], Polly A. Newcomb[5,51], David Duggan[52], Jeremy P. Cheadle[53], Richard Kaplan[54], Rachel Kerr[55], David Kerr[56], Iva Kirac[57], Jan Böhm[58], Jukka-Pekka Mecklin[59], Pekka Jousilahti[60], Paul Knekt[60], Lauri A. Aaltonen[61,62], Harri Rissanen[63], Eero Pukkala[64,65], Johan G. Eriksson[66,67,68], Tatiana Cajuso[61,62], Ulrika Hänninen[61,62], Johanna Kondelin[61,62], Kimmo Palin[61,62], Tomas Tanskanen[61,62], Laura Renkonen-Sinisalo[69], Satu Männistö[63], Demetrius Albanes[70], Stephanie J. Weinstein[70], Edward Ruiz-Narvaez[71], Julie R. Palmer[72,73], Daniel D. Buchanan[74,75,76], Elizabeth A. Platz[77], Kala Visvanathan[77], Cornelia M. Ulrich[78], Erin Siegel[79], Stefanie Brezina[80], Andrea Gsur[80], Peter T. Campbell[81], Jenny Chang-Claude[82,83], Michael Hoffmeister[84], Hermann Brenner[84,85,86], Martha L. Slattery[87], John D. Potter[5,88], Kostas K. Tsilidis[89,90], Matthias B. Schulze[91,92], Marc J. Gunter[93], Neil Murphy[93], Antoni Castells[94], Sergi Castellví-Bel[94], Leticia Moreira[94], Volker Arndt[84], Anna Shcherbina[95], D. Timothy Bishop[96], Graham G. Giles[44,97,98], Melissa C. Southey[97,98,99], Gregory E. Idos[100], Kevin J. McDonnell[25,27,100], Zomoroda Abu-Ful[26], Joel K. Greenson[25,27,101], Katerina Shulman[26], Flavio Lejbkowicz[25,26,102], Kenneth Offit[103,104], Yu-Ru Su[105], Robert Steinfelder[5], Temitope O. Keku[106], Bethany van Guelpen[107,108], Thomas J. Hudson[109], Heather Hampel[110], Rachel Pearlman[110], Sonja I. Berndt[70], Richard B. Hayes[111], Marie Elena Martinez[112,113], Sushma S. Thomas[114], Paul D. P. Pharoah[115], Susanna C. Larsson[116], Yun Yen[117], Heinz-Josef Lenz[118], Emily White[5,119], Li Li[22], Kimberly F. Doheny[120], Elizabeth Pugh[120], Tameka Shelford[120], Andrew T. Chan[121,122,123,124,125,126], Marcia Cruz-Correa[127], Annika Lindblom[128,129], David J. Hunter[124,130], Amit D. Joshi[123,124], Clemens Schafmayer[131], Peter C. Scacheri[132], Anshul Kundaje[95,133], Robert E. Schoen[134], Jochen Hampe[135], Zsofia K. Stadler[104,136], Pavel Vodicka[137,138,139], Ludmila Vodickova[137,138,139], Veronika Vymetalkova[137,138,139], Christopher K. Edlund[21], W. James Gauderman[21], David Shibata[140], Amanda Toland[141], Sanford Markowitz[142], Andre Kim[21], Stephen J. Chanock[70], Franzel van Duijnhoven[143], Edith J. M. Feskens[144], Lori C. Sakoda[5,145], Manuela Gago-Dominguez[146,147], Alicja Wolk[116], Barbara Pardini[148,149], Liesel M. FitzGerald[97,150], Soo Chin Lee[151], Shuji Ogino[124,152,153,154], Stephanie A. Bien[5], Charles Kooperberg[5], Christopher I. Li[5], Yi Lin[5], Ross Prentice[5,155], Conghui Qu[5], Stéphane Bézieau[156], Taiki Yamaji[157], Norie Sawada[158], Motoki Iwasaki[157,158], Loic Le Marchand[159], Anna H. Wu[160], Chenxu Qu[161], Caroline E. McNeil[161], Gerhard Coetzee[162], Caroline Hayward[163], Ian J. Deary[164], Sarah E. Harris[164], Evropi Theodoratou[165], Stuart Reid[10], Marion Walker[10], Li Yin Ooi[10,166], Ken S. Lau[167], Hongyu Zhao[9,168,169], Li Hsu[5,170], Qiuyin Cai[1], Malcolm G. Dunlop[10], Stephen B. Gruber[100], Richard S. Houlston[6], Victor Moreno[14,15,16,17], Graham Casey[18], Ulrike Peters[5,171], Ian Tomlinson[7,19] & Wei Zheng[1] ✉

[1]Division of Epidemiology, Department of Medicine, Vanderbilt-Ingram Cancer Center, Vanderbilt Epidemiology Center, Vanderbilt University Medical Center, Nashville, TN, USA. [2]Department of Biomedical Informatics, Vanderbilt University School of Medicine, Nashville, TN, USA. [3]Department of Biostatistics, Vanderbilt University Medical Center, Nashville, TN, USA. [4]Vanderbilt Genetics Institute, Vanderbilt University Medical Center, Nashville 37232 TN, USA. [5]Public Health Sciences Division, Fred Hutchinson Cancer Center, Seattle, WA, USA. [6]Division of Genetics and Epidemiology, Institute of Cancer Research, London, UK. [7]Edinburgh Cancer Research Centre, Institute of Genomics and Cancer, University of Edinburgh, Edinburgh, UK. [8]Genomic Medicine Group, Instituto de Investigacion Sanitaria de Santiago, Santiago de Compostela, Spain. [9]Department of Biostatistics, Yale School of Public Health, New Haven, CT, USA. [10]Colon Cancer Genetics Group, Medical Research Council Human Genetics Unit, Institute of Genetics and Cancer, University of Edinburgh, Edinburgh, UK. [11]Danish Institute for Advanced Study, Department of Public Health, University of Southern Denmark, Odense, Denmark. [12]Genomic Medicine Institute, Cleveland Clinic, Cleveland, OH, USA. [13]Population and Cancer Prevention Program, Case Comprehensive Cancer Center, Cleveland, OH, USA. [14]Colorectal Cancer Group, ONCOBELL Program, Bellvitge Biomedical Research Institute, Barcelona, Spain. [15]Consortium for Biomedical Research in Epidemiology and Public Health, Madrid, Spain. [16]Department of Clinical Sciences, Faculty of Medicine, University of Barcelona, Barcelona, Spain. [17]Oncology Data Analytics Program, Catalan Institute of Oncology, Barcelona, Spain. [18]Center for Public Health Genomics, Department of Public Health Sciences, University of Virginia, Charlottesville, VA, USA. [19]Institute of Cancer and Genomic Sciences, College of Medical and Dental Sciences, University of Birmingham, Birmingham, UK. [20]Department of Public Health, Richard Doll Building, University of Oxford, Oxford, UK. [21]Department of Preventive Medicine, USC Norris Comprehensive Cancer Center, Keck School of Medicine, University of Southern California, Los Angeles, CA, USA. [22]Case Comprehensive Cancer Center, Case Western Reserve University, Cleveland, OH, USA. [23]Department of Population and Quantitative Health Sciences, Case Western Reserve University, Cleveland, OH, USA. [24]The Steve and Cindy Rasmussen Institute for Genomic Medicine, Nationwide Children's Hospital, Columbus, OH, USA.

[25]Clalit National Cancer Control Center, Haifa, Israel. [26]Department of Community Medicine and Epidemiology, Lady Davis Carmel Medical Center, Haifa, Israel. [27]Ruth and Bruce Rappaport Faculty of Medicine, Technion-Israel Institute of Technology, Haifa, Israel. [28]Biomedicine Institute, University of León, León, Spain. [29]Center for Colorectal Cancer, National Cancer Center Hospital, National Cancer Center, Gyeonggi-do, South Korea. [30]Department of Cancer Biomedical Science, Graduate School of Cancer Science and Policy, National Cancer Center, Gyeonggi-do, South Korea. [31]Department of Epidemiology and Health Promotion, Graduate School of Public Health, Yonsei University, Seoul, South Korea. [32]Department of Preventive Medicine, Chonnam National University Medical School, Gwangju, South Korea. [33]Cancer Research Institute, Seoul National University, Seoul, South Korea. [34]Department of Preventive Medicine, Seoul National University College of Medicine, Seoul, South Korea. [35]Department of Social and Preventive Medicine, Hallym University College of Medicine, Okcheon-dong, South Korea. [36]Division of Cancer Epidemiology and Prevention, Aichi Cancer Center Research Institute, Nagoya, Japan. [37]Department of Epidemiology, Nagoya University Graduate School of Medicine, Nagoya, Japan. [38]Division of Molecular and Clinical Epidemiology, Aichi Cancer Center Research Institute, Nagoya, Japan. [39]Laboratory of Clinical Genome Sequencing, Department of Computational Biology and Medical Sciences, Graduate School of Frontier Sciences, University of Tokyo, Tokyo, Japan. [40]Laboratory of Genome Technology, Human Genome Center, Institute of Medical Science, University of Tokyo, Tokyo, Japan. [41]School of Public Health, Sun Yat-sen University, Guangzhou, China. [42]State Key Laboratory of Oncogenes and Related Genes and Department of Epidemiology, Shanghai Cancer Institute, Renji Hospital, Shanghai Jiaotong University School of Medicine, Shanghai, China. [43]State Key Laboratory of Oncology in South China, Cancer Center, Sun Yat-sen University, Guangzhou, China. [44]Centre for Epidemiology and Biostatistics, Melbourne School of Population and Global Health, University of Melbourne, Melbourne, VIC, Australia. [45]Department of Epidemiology, School of Public Health and Institute of Health and Environment, Seoul National University, Seoul, South Korea. [46]Department of Laboratory Medicine and Pathology, Mayo Clinic Arizona, Scottsdale, AZ, USA. [47]Department of Medicine, Samuel Oschin Comprehensive Cancer Institute, Cedars-Sinai Medical Center, Los Angeles, CA, USA. [48]Division of Oncology, Department of Medicine, Cedars-Sinai Cancer Research Center for Health Equity, Los Angeles, CA, USA. [49]Lunenfeld Tanenbaum Research Institute, Mount Sinai Hospital, University of Toronto, Toronto, ON, Canada. [50]Division of Biomedical Sciences, Memorial University of Newfoundland, St. John, ON, Canada. [51]School of Public Health, University of Washington, Seattle, WA, USA. [52]City of Hope National Medical Center, Translational Genomics Research Institute, Phoenix, AZ, USA. [53]Institute of Medical Genetics, Cardiff University, Cardiff, UK. [54]MRC Clinical Trials Unit, Medical Research Council, Cardiff, UK. [55]Department of Oncology, University of Oxford, Oxford, UK. [56]Radcliffe Department of Medicine, University of Oxford, Oxford, UK. [57]Department of Surgical Oncology, University Hospital for Tumors, Sestre milosrdnice University Hospital Center, Zagreb, Croatia. [58]Department of Pathology, Central Finland Health Care District, Jyväskylä, Finland. [59]Central Finland Health Care District, Jyväskylä, Finland. [60]Department of Health and Welfare, Finnish Institute for Health and Welfare, Helsinki, Finland. [61]Department of Medical and Clinical Genetics, University of Helsinki, Helsinki, Finland. [62]Genome-Scale Biology Research Program, University of Helsinki, Helsinki, Finland. [63]Department of Public Health and Welfare, Finnish Institute for Health and Welfare, Helsinki, Finland. [64]Faculty of Social Sciences, Tampere University, Tampere, Finland. [65]Finnish Cancer Registry, Institute for Statistical and Epidemiological Cancer Research, Helsinki, Finland. [66]Folkhälsan Research Centre, University of Helsinki, Helsinki, Finland. [67]Human Potential Translational Research Programme, National University of Singapore, Singapore, Singapore. [68]Unit of General Practice and Primary Health Care, University of Helsinki and Helsinki University Hospital, Helsinki, Finland. [69]Department of Surgery, Abdominal Centre, Helsinki University Hospital, Helsinki, Finland. [70]Division of Cancer Epidemiology and Genetics, National Cancer Institute, National Institutes of Health, Bethesda, MD, USA. [71]Department of Nutritional Sciences, School of Public Health, University of Michigan, Ann Arbor, MI, USA. [72]Department of Medicine, Boston University School of Medicine, Boston, MA, USA. [73]Slone Epidemiology Center at Boston University, Boston, MA, USA. [74]Colorectal Oncogenomics Group, Department of Clinical Pathology, University of Melbourne, Parkville, VIC, Australia. [75]Genomic Medicine and Family Cancer Clinic, Royal Melbourne Hospital, Parkville, VIC, Australia. [76]University of Melbourne Centre for Cancer Research, Victorian Comprehensive Cancer Centre, Parkville, VIC, Australia. [77]Department of Epidemiology, Johns Hopkins Bloomberg School of Public Health, Baltimore, MD, USA. [78]Huntsman Cancer Institute and Department of Population Health Sciences, University of Utah, Salt Lake City, UT, USA. [79]Cancer Epidemiology Program, H. Lee Moffitt Cancer Center and Research Institute, Tampa, FL, USA. [80]Institute of Cancer Research, Department of Medicine I, Medical University Vienna, Vienna, Austria. [81]Department of Epidemiology and Population Health, Albert Einstein College of Medicine, New York, NY, USA. [82]Division of Cancer Epidemiology, German Cancer Research Center, Heidelberg, Germany. [83]University Medical Centre Hamburg-Eppendorf, University Cancer Centre Hamburg, Hamburg, Germany. [84]Division of Clinical Epidemiology and Aging Research, German Cancer Research Center, Heidelberg, Germany. [85]Division of Preventive Oncology, German Cancer Research Center and National Center for Tumor Diseases, Heidelberg, Germany. [86]German Cancer Consortium, German Cancer Research Center, Heidelberg, Germany. [87]Department of Internal Medicine, University of Utah, Salt Lake City, UT, USA. [88]Research Centre for Hauora and Health, Massey University, Wellington, New Zealand. [89]Department of Epidemiology and Biostatistics, School of Public Health, Imperial College London, London, UK. [90]Department of Hygiene and Epidemiology, University of Ioannina School of Medicine, Ioannina, Greece. [91]Department of Molecular Epidemiology, German Institute of Human Nutrition Potsdam-Rehbruecke, Nuthetal, Germany. [92]Institute of Nutritional Science, University of Potsdam, Potsdam, Germany. [93]Nutrition and Metabolism Branch, International Agency for Research on Cancer, World Health Organization, Lyon, France. [94]Gastroenterology Department, Hospital Clínic, Institut d'Investigacions Biomèdiques August Pi i Sunyer, Centro de Investigación Biomédica en Red de Enfermedades Hepáticas y Digestivas, University of Barcelona, Barcelona, Spain. [95]Department of Genetics, Stanford University, Stanford, CA, USA. [96]Leeds Institute of Medical Research at St James's, University of Leeds, Leeds, UK. [97]Cancer Epidemiology Division, Cancer Council Victoria, Melbourne, VIC, Australia. [98]Precision Medicine, School of Clinical Sciences at Monash Health, Monash University, Clayton, VIC, Australia. [99]Department of Clinical Pathology, University of Melbourne, Melbourne, VIC, Australia. [100]Department of Medical Oncology and Center For Precision Medicine, City of Hope National Medical Center, Duarte, CA, USA. [101]Department of Pathology, University of Michigan, Ann Arbor, MI, USA. [102]Clalit Health Services, Personalized Genomic Service, Lady Davis Carmel Medical Center, Haifa, Israel. [103]Clinical Genetics Service, Department of Medicine, Memorial Sloan-Kettering Cancer Center, New York, NY, USA. [104]Department of Medicine, Weill Cornell Medical College, New York, NY, USA. [105]Kaiser Permanente Washington Health Research Institute, Seattle, WA, USA. [106]Center for Gastrointestinal Biology and Disease, University of North Carolina, Chapel Hill, NC, USA. [107]Department of Radiation Sciences, Oncology Unit, Umeå University, Umeå, Sweden. [108]Wallenberg Centre for Molecular Medicine, Umeå University, Umeå, Sweden. [109]Ontario Institute for Cancer Research, Toronto, ON, Canada. [110]Division of Human Genetics, Department of Internal Medicine, Ohio State University Comprehensive Cancer Center, Columbus, OH, USA. [111]Division of Epidemiology, Department of Population Health, New York University School of Medicine, New York, NY, USA. [112]Department of Family Medicine and Public Health, University of California San Diego, La Jolla, CA, USA. [113]Population Sciences, Disparities and Community Engagement, University of California San Diego Moores Cancer Center, La Jolla, CA, USA. [114]Fred Hutchinson Cancer Center, Seattle, WA, USA. [115]Department of Public Health and Primary Care, University of Cambridge, Cambridge, UK. [116]Institute of Environmental Medicine, Karolinska Institutet, Stockholm, Sweden. [117]Taipei Medical University, Taipei, Taiwan. [118]Department of Medicine, Keck School of Medicine, University of Southern California, Los Angeles, CA, USA. [119]Department of Epidemiology, University of Washington School of Public Health, Seattle, WA, USA. [120]Center for Inherited Disease Research, Department of Genetic Medicine, Johns Hopkins University School of Medicine, Baltimore, MD, USA. [121]Broad Institute of Harvard and MIT, Cambridge, MA, USA. [122]Channing Division of Network Medicine, Brigham and Women's Hospital and Harvard Medical

School, Boston, MA, USA. [123]Clinical and Translational Epidemiology Unit, Massachusetts General Hospital and Harvard Medical School, Boston, MA, USA. [124]Department of Epidemiology, Harvard T.H. Chan School of Public Health, Harvard University, Boston, MA, USA. [125]Department of Immunology and Infectious Diseases, Harvard T.H. Chan School of Public Health, Harvard University, Boston, MA, USA. [126]Division of Gastroenterology, Massachusetts General Hospital and Harvard Medical School, Boston, MA, USA. [127]Comprehensive Cancer Center, University of Puerto Rico, San Juan, Puerto Rico. [128]Department of Clinical Genetics, Karolinska University Hospital, Stockholm, Sweden. [129]Department of Molecular Medicine and Surgery, Karolinska Institutet, Stockholm, Sweden. [130]Nuffield Department of Population Health, University of Oxford, Oxford, UK. [131]Department of General Surgery, University Hospital Rostock, Rostock, Germany. [132]Department of Genetics and Genome Sciences, Case Western Reserve University, Cleveland, OH, USA. [133]Department of Computer Science, Stanford University, Stanford, CA, USA. [134]Department of Medicine and Epidemiology, University of Pittsburgh Medical Center, Pittsburgh, PA, USA. [135]Department of Medicine I, University Hospital Dresden, Technische Universität Dresden, Dresden, Germany. [136]Department of Medicine, Memorial Sloan-Kettering Cancer Center, New York, NY, USA. [137]Department of Molecular Biology of Cancer, Institute of Experimental Medicine of the Czech Academy of Sciences, Prague, Czech Republic. [138]Faculty of Medicine and Biomedical Center in Pilsen, Charles University, Pilsen, Czech Republic. [139]Institute of Biology and Medical Genetics, First Faculty of Medicine, Charles University, Prague, Czech Republic. [140]Department of Surgery, University of Tennessee Health Science Center, Memphis, TN, USA. [141]Departments of Cancer Biology and Genetics and Internal Medicine, Comprehensive Cancer Center, Ohio State University, Columbus, OH, USA. [142]Departments of Medicine and Genetics, Case Comprehensive Cancer Center, Case Western Reserve University and University Hospitals of Cleveland, Cleveland, OH, USA. [143]Division of Human Nutrition and Health, Wageningen University and Research, Wageningen, The Netherlands. [144]Division of Human Nutrition, Wageningen University and Research, Wageningen, The Netherlands. [145]Division of Research, Kaiser Permanente Northern California, Oakland, CA, USA. [146]Genomic Medicine Group, Galician Public Foundation of Genomic Medicine, Servicio Galego de Saude, Santiago de Compostela, Spain. [147]Instituto de Investigación Sanitaria de Santiago de Compostela, Santiago de Compostela, Spain. [148]Candiolo Cancer Institute FPO-IRCCS, Candiolo, (TO), Italy. [149]Italian Institute for Genomic Medicine, Candiolo Cancer Institute FPO-IRCCS, Candiolo, (TO), Italy. [150]Menzies Institute for Medical Research, University of Tasmania, Hobart, TAS, Australia. [151]National University Cancer Institute, Singapore, Cancer Science Institute of Singapore, National University of Singapore, Singapore, Singapore. [152]Broad Institute of MIT and Harvard, Cambridge, MA, USA. [153]Cancer Immunology Program, Dana-Farber Harvard Cancer Center, Boston, MA, USA. [154]Department of Pathology, Brigham and Women's Hospital, Harvard Medical School, Boston, MA, USA. [155]Department of Biostatistics, University of Washington, Seattle, WA, USA. [156]Service de Génétique Médicale, Centre Hospitalier Universitaire Nantes, Nantes, France. [157]Division of Epidemiology, National Cancer Center Institute for Cancer Control, National Cancer Center, Tokyo, Japan. [158]Division of Cohort Research, National Cancer Center Institute for Cancer Control, National Cancer Center, Tokyo, Japan. [159]Cancer Center, University of Hawaii, Honolulu, HI, USA. [160]Preventative Medicine, University of Southern California, Los Angeles, CA, USA. [161]USC Norris Comprehensive Cancer Center, Keck School of Medicine, University of Southern California, Los Angeles, CA, USA. [162]Van Andel Research Institute, Grand Rapids, MI, USA. [163]MRC Human Genetics Unit, Institute of Genomics and Cancer, University of Edinburgh, Edinburgh, UK. [164]Lothian Birth Cohorts group, Department of Psychology, University of Edinburgh, Edinburgh, UK. [165]Centre for Global Health, Usher Institute, University of Edinburgh, Edinburgh, UK. [166]Department of Pathology, National University Hospital, National University Health System, Singapore, Singapore. [167]Epithelial Biology Center and Department of Cell and Developmental Biology, Vanderbilt University School of Medicine, Nashville, TN, USA. [168]Department of Genetics, Yale School of Medicine, New Haven, CT, USA. [169]Program in Computational Biology and Bioinformatics, Yale University, New Haven, CT, USA. [170]Department of Biostatistics, School of Public Health, University of Washington, Seattle, WA, USA. [171]Department of Epidemiology, University of Washington, Seattle, WA, USA. ✉e-mail: wei.zheng@vanderbilt.edu

