## [Peer Review File · Nature Communications]

Fine-mapping analysis including over 254,000 East Asian and European descendants identifies 136 putative colorectal cancer susceptibility genesREVIEWER COMMENTS

Reviewer #1 (Remarks to the Author):

The study employs a comprehensive fine-mapping and functional annotation approach with a large cohort to investigate CRC risk. Authors identify 238 independent association signals and credible causal variants. Functional exploration, including transcriptome and methylome data, reveals 136 putative CRC susceptibility genes, 56 of which are novel. Single-cell RNA-seq data and whole exome sequencing data was also included. The study extensively employs in silico data for fine-mapping and functional annotation, lacking experimental validation. While miRNA genes and pseudogenes are identified, there is no in-depth analysis. Some clarifications are needed:

(1) A conditional $P < 1 \times 10^{-6}$ threshold is chosen, deviating from the common GWAS threshold of 10^{-8} . This may contribute to the increased number of independent association signals. Comparisons with the original GWAS, particularly with the 47 newly identified signals, warrant exploration by lowering the GWAS threshold.

(2) The identification of 198 and 45 independent association signals in European and East Asian descendants, respectively, requires further discussion. How does this compare to the original GWAS? Are these population-specific or due to the smaller East Asian sample size? The differences between East Asian and European populations and the reliability of combined (trans-ancestry) analysis need clarification.

(3) For the identification of CCV in trans-ancestry analysis, how the LD was estimated? Wouldn't it be useful to consider LD blocks from other available population to reduce the number of CCV rather than using it just from two populations?

(4) The use of different minor allele frequency (MAF) thresholds (e.g., 0.005, 0.1, and 0.5) needs normalization or explanation across analyses.

(5) The overlap with previous eQTL and mQTL studies isn't clearly stated.

(6) The significance and power of differential expression analysis in specific cell types require clarification.

Minor comment: The mapping of 136 credible target genes to 126 pathways, with overlapping pathways and multiple tools, should be presented more distinctly. Separating tools and highlighting common pathways would enhance clarity.

Reviewer #2 (Remarks to the Author):

This paper begins with fine-mapping of a previous GWAS conducted by the same group (PMID: 36539618 Fernandez-Rozadilla et al, 2022). The methods are appropriate and appear to be based on approaches used by the Breast Cancer Consortium to fine-map and identify target gene of breast cancer GWAS. This paper has refined the target gene predictions for their 2022 paper and outlined the evidence associated with each prediction. This will become an important reference set for colorectal cancer researchers. Pathway analyses were re-performed with the more refined predictions which confirmed previous enriched pathways known to play a role in colorectal cancer development.

Of note, a conditional analyses was performed in Fernandez-R et al . In regards to the fine-mapping performed in this paper, what was done different in this study.

“13 independent new risk SNPs in conditional analysis” were identified in Fernandez-R et al. Is there overlap between these 13 SNPs and the SNPs identified in this paper?

eQTLs that did not colocalize with GWAS signals should not be considered evidence of “putative” or “credible” target genes. Please confirm that this was the case.

For Line 459 “9 genes as putative targets of eight signals “. Please confirm what you mean by putative vs credible.

Line 444. Of the “84 genes from the mQTL analysis”, how many are supported by eQTL colocalization? It is not clear why an association with methylation without expression would be evidence for a target gene? In these cases, it is possible that the variant effects

methylation but the methylation has no functional significance on the gene. Is mQTL/GWAS colocalization considered to be supportive evidence in Figure 3 and 4?. If so, please justify.

The results sections are brief. Important experimental details are required in the results section so that the reader doesn't have to constantly refer to the methods.

Examples.

1. How were the sets of CCVs defined.
2. eQTL analyses – what tissue type was the transcriptome data derived from?
3. What is meant by functional genomic evidence in Line 450 “Of these, 45 (21.9%) genes were supported by the functional genomic evidence.” Is this the target gene prediction from INQUISIT??
4. Line 470 – Please define ABS.
5. In Figure 3 and 4. Please define which colocalization analyses were considered evidence.

Authors' Responses to Reviewers' Comments

Reviewer #1 (Remarks to the Author):

The study employs a comprehensive fine-mapping and functional annotation approach with a large cohort to investigate CRC risk. Authors identify 238 independent association signals and credible causal variants. Functional exploration, including transcriptome and methylome data, reveals 136 putative CRC susceptibility genes, 56 of which are novel. Single-cell RNA-seq data and whole exome sequencing data was also included. The study extensively employs in silico data for fine-mapping and functional annotation, lacking experimental validation. While miRNA genes and pseudogenes are identified, there is no in-depth analysis. Some clarifications are needed:

(1) A conditional $P < 1 \times 10^{-6}$ threshold is chosen, deviating from the common GWAS threshold of 10^{-8} . This may contribute to the increased number of independent association signals. Comparisons with the original GWAS, particularly with the 47 newly identified signals, warrant exploration by lowering the GWAS threshold.

Response: We appreciate the reviewer's comments. In this study, we aimed to identify secondary independent association signals within each risk locus. Because this is not a search for novel association signals across the genome, it would be too stringent to use the significance threshold of 5×10^{-8} that is conventionally used for genome-wide association studies. Indeed, it has been shown that using p-value of 5×10^{-8} in fine-mapping analyses can miss many secondary independent signals (PMID: 33978749). We selected 1×10^{-6} as the significance threshold in our study primarily based on a large fine-mapping study conducted for breast cancer (PMID: 31911677). In that study, a different significant threshold was tested initially (1×10^{-4} , 1×10^{-5} , 1×10^{-6} , 1×10^{-7} , and 1×10^{-8}), and a threshold of 1×10^{-6} was finally used, as it minimizes both Type 1 and 2 errors. This threshold was used in multiple subsequent fine-mapping analyses. We have clarified this issue in the manuscript, as shown below:

In page 15, paragraph 2: *"We considered the threshold of conditional $P < 1 \times 10^{-6}$ to determine independent significant associations to balance both Type 1 and 2 errors, as recommended by a previous fine-mapping study in breast cancer¹²."*

(2) The identification of 198 and 45 independent association signals in European and East Asian descendants, respectively, requires further discussion. How does this compare to the original GWAS? Are these population-specific or due to the smaller East Asian sample size? The differences between East Asian and European populations and the reliability of combined (trans-ancestry) analysis need clarification.

Response: We thank the reviewer for their comments. In the Methods section of the revised manuscript, we included a description comparing the independent association signals identified in population-specific analyses with those identified in trans-ancestry analysis and previous GWAS. We believe that the difference in the number of independent association signals identified in these two analyses is largely due to the difference in study sample size. We also added some discussions regarding population-specific association signals in the revised manuscript, as shown below:

In page 27, paragraph 3: “For independent association signals identified in ancestry-specific analyses, we compared them with those from trans-ancestry analyses by assessing correlations between their lead variants within each risk region. If a signal was consistently found in both ancestry-specific and trans-ancestry analyses (i.e., the same lead variant or correlated lead variants with LD $r^2 > 0.1$ in each corresponding population), we considered it as a sharing signal between Asian and European-ancestry populations. Otherwise, they were defined as ancestry-specific signals.”

In pages 24-25: “The trans-ancestry and ancestry-specific fine-mapping analyses conducted in this study not only enabled the discovery of independent association signals that are shared across populations of European and East Asian ancestry, but also revealed ancestry-specific signals. The larger sample size of the European-ancestry study enabled us to identify a larger number of independent association signals than the study conducted in Asians. However, there are some ancestry-specific signals identified in this study, which is most likely due to differences in LD structures and allele frequency between these two populations. Indeed, we observed distinct differences in the allele frequency for most ancestry-specific signals, as shown in **Supplementary Tables 4 and 5**. For instance, the lead variant of 24 European ancestry-specific signals (40%, 24/60) is not detected among East Asian-ancestry populations. On the other hand, fine-mapping analyses capitalizing on ancestry differences in LD structure can substantially reduce the credible set size compared to European ancestry-specific analysis. This highlights the value of multi-ancestry fine-mapping over single-ancestry analysis. Our analysis is limited to two ancestry groups. Further studies should increase the diversity of genetic data, including those from other racial groups.”

(3) For the identification of CCV in trans-ancestry analysis, how the LD was estimated? Would’nt it be useful to consider LD blocks from other available population to reduce the number of CCV rather than using it just from two populations?

Response: In our trans-ancestry analysis, to account for differences in the LD structure, we conducted conditional analysis in each population and then meta-analyzed conditioned results using the fixed-effects inverse variance weighted model with METAL. We used genotyping data from 6,684 unrelated samples of Asian descent and from 503 European samples in the 1000 Genome project as the reference for LD estimation. To identify CCVs for each independent signal, the conditional analyses were conducted by adjusting for the lead variants of the other signals from the trans-ancestral summary statistics within the same risk region. We included detailed descriptions for the conditional analyses in trans-ancestry in the revised manuscript. We agree with the reviewer that it would be helpful to add other populations in the fine-mapping analyses. However, currently, we only have data from two populations and acknowledge that this is a limitation of our study.

In page 15, paragraph 2: “We used forward stepwise conditional analyses to identify independent association signals in each region in each population, conditioning on the most significant association signal from the trans-ancestral summary statistics (**Supplementary Figure 1, Methods**). We then meta-analyzed the conditioned data using the fixed-effects inverse variance weighted model.”

In pages 16, paragraph 3: “To identify CCVs for each independent association signal, we conducted conditional analysis with adjustment of the lead variants for other signals in the same risk region. We conducted this analysis for trans-ancestral independent signals separately for each population to account for differences in the LD structure and then combined conditioned results.”

In page 25, paragraph 1: “On the other hand, fine-mapping analyses capitalizing on ancestry differences in LD structure can substantially reduce the credible set size compared to European-ancestry specific

analysis. This highlights the value of multi-ancestry fine-mapping over single-ancestry analysis. Our analysis is limited to two ancestry groups. Further studies should increase the diversity of genetic data, including those from other racial groups.”

(4) The use of different minor allele frequency (MAF) thresholds (e.g., 0.005, 0.1, and 0.5) needs normalization or explanation across analyses.

Response: We thank the reviewer for the comment. Two MAF thresholds (0.005 and 0.01) were mentioned in our original manuscript. The threshold of MAF=0.005 was used in the previous GWAS paper, while MAF=0.01 was used in our study. To avoid confusion, we rephrased the description regarding GWAS datasets and removed the threshold of 0.005 in the revised manuscript.

In pages 25-26: “*GWAS data and meta-analysis: The GWAS data used in this study comprised 100,204 CRC cases and 154,587 controls (Supplementary Table 1), which were grouped into 31 GWAS analytical units based on study or genotyping platform as consistent with the original reports. Of them, 17 datasets were derived from populations of European descent and 14 were from populations of Asian descent. These 31 GWAS datasets were meta-analyzed under the fixed-effects inverse variance weighted model implemented in METAL³⁰. Further details regarding each analytical unit and meta-analysis were described previously⁷.*”

(5) The overlap with previous eQTL and mQTL studies isn't clearly stated.

Response: We compared genes identified both in our eQTL and mQTL and previous studies and added the results to the revised manuscript.

In page 18, paragraph 2: “*At Bonferroni-corrected $P < 0.05$, we identified 153 genes associated with the lead variants, including 127 genes in 65 independent association signals and 30 genes in 15 signals identified from trans-ancestry and European-ancestry specific analyses, respectively. We also identified the PPP1R21 gene in an Asian-specific risk signal (lead variant rs77272589) (Supplementary Table 13). Out of these 153 genes, 37 had been previously identified by eQTL analysis^{5,10,11}”*

In page 19, paragraph 1: “*We found that DNA methylation levels at CpG sites for 84 genes were associated with 71 independent association signals, including 14 genes identified in previous mQTL analysis¹¹(Supplementary Table 14).*”

(6) The significance and power of differential expression analysis in specific cell types require clarification.

Response: In the revised manuscript, we added the significant threshold to determine genes with significantly differential expression between cell types in single-cell RNA-seq analysis.

In page 36, paragraph 1: “*The criteria $|\log_2 \text{fold change (FC)}| > 1$ and $P < 0.05$ were applied to determine genes with significantly differential expression between cell types.*”

Minor comment: The mapping of 136 credible target genes to 126 pathways, with overlapping pathways and multiple tools, should be presented more distinctly. Separating tools and highlighting common pathways would enhance clarity.

Response: We thank the reviewer for the comments. We reorganized the results of the pathway analysis according to pathway databases and highlighted the common pathways by colors. The updated results are exhibited in Supplementary Table 22 in the revised **Supplementary Tables** file.

Reviewer #2 (Remarks to the Author):

This paper begins with fine-mapping of a previous GWAS conducted by the same group (PMID: 36539618 Fernandez-Rozadilla et al, 2022). The methods are appropriate and appear to be based on approaches used by the Breast Cancer Consortium to fine-map and identify target gene of breast cancer GWAS. This paper has refined the target gene predictions for their 2022 paper and outlined the evidence associated with each prediction. This will become an important reference set for colorectal cancer researchers. Pathway analyses were re-performed with the more refined predictions which confirmed previous enriched pathways known to play a role in colorectal cancer development.

Of note, a conditional analyses was performed in Fernandez-R et al . In regards to the fine-mapping performed in this paper, what was done different in this study.

“13 independent new risk SNPs in conditional analysis” were identified in Fernandez-R et al. Is there overlap between these 13 SNPs and the SNPs identified in this paper?

Response: We thank the reviewer for the positive comments on our work. The major purpose of our previous GWAS (PMID: 36539618, Fernandez-Rozadilla et al, 2022) was to identify novel risk loci for CRC, while the major purpose of this current fine-mapping analysis is to identify putative causal variants and genes. Therefore, different strategies were used in these two studies. In the paper by Fernandez-Rozadilla et al, we performed conditional analyses within 1 Mb of each CRC risk loci conditioning on only index SNP, and we did not call it a fine-mapping analysis. In the current study, however, we performed forward stepwise conditional analyses in each of the 142 GWAS-identified regions for CRC risk. As a result, we identified 47 additional association signals that were not reported previously. **More importantly, the conditional analysis was just to set the stage for multiple subsequent analyses we performed to identify putative causal variants and genes and biological pathways involved in CRC development.**

eQTLs that did not colocalize with GWAS signals should not be considered evidence of “putative” or “credible” target genes. Please confirm that this was the case.

Response: We have carefully checked the terms “putative” and “credible” throughout the manuscript and removed “putative” for “target genes” in the context of QTL analysis without colocalization evidence.

For Line 459 “9 genes as putative targets of eight signals “. Please confirm what you mean by putative vs credible.

Response: The new sentence reads like this:

In page 20, paragraph 1: “Among them, 56 genes were newly identified as potential targets for CRC risk associations, including nine genes in eight novel association signals in this study (**Figure 3**)”.

Line 444. Of the “84 genes from the mQTL analysis”, how many are supported by eQTL colocalization? It is not clear why an association with methylation without expression would be evidence for a target gene? In these cases, it is possible that the variant affects methylation but the methylation has no functional significance on the gene. Is mQTL/GWAS colocalization considered to be supportive evidence in Figure 3 and 4?. If so, please justify.

Response: We thank the reviewer for the comments. In our revised manuscript, we compared genes identified in mQTL and eQTL analyses. Our result show that 34% (29/84) genes from the mQTL analysis were also identified in the eQTL analysis, sharing the same lead variant. This fraction is not large, which is likely due to a smaller sample size in the mQTL analysis (n=321) when compared with the eQTL analysis (n=1299). Nevertheless, our result is in line with previous observations showing about 27% of GWAS hits colocalizing with both mQTL and eQTL in the same tissue (PMID: 36510025).

Recent studies provided strong evidence that mQTLs explain large fractions of GWAS-identified signals (PMID: 36510025, PMID: 37601976) and mQTLs are strongly enriched in distal enhancers and insulators, which are important for gene regulation (PMID: 36510025). Furthermore, the link between DNA methylation and GWAS signal involves additional molecular phenotypes other than gene expression. A recent study showed that some mQTLs are also associated with other molecular phenotypes, such as histone modification and chromatin accessibility. Notably, 29 of the 55 genes with evidence of mQTL but no eQTL in our study were supported by other layers of evidence, including functional genomic data, and their associations with CRC risk through TWAS and eQTL colocalization. We included the results in the revised manuscript. We also updated Figures 3 and 4.

“Of these, 29 genes were identified in both mQTL and eQTL analyses, and 45 (21.9%) genes were also identified as targets of CCVs in the in-silico analyses based on functional genomic data as described above.” (page 19, paragraph 2)

In page 19, paragraph 2: “Of these, 45 (21.9%) genes were also identified as targets of CCVs by in-silico analyses based on functional genomic data as described above, and 29 genes were identified in both mQTL and eQTL analyses that is in line with previous observations in the overlap fraction between mQTL and eQTL¹⁴. We considered genes with evidence of only mQTL colocalization, as the enrichment of mQTLs in gene regulatory elements, as well as their implications in other molecular phenotypes, such as chromatin accessibility^{14,15}. Notably, of the 55 genes only identified in the mQTL analysis, seven genes were supported by the above in silico analyses with functional genomic data, and 22 genes showed association with CRC risk in previous TWAS and eQTL colocalization analysis^{7,11,16,17}.”

The results sections are brief. Important experimental details are required in the results section so that the reader doesn't have to constantly refer to the methods.

Response: We thank the reviewer for the comments. We added details of the methods to the Results section in the revised manuscript as detailed below.

Examples.

1. How were the sets of CCVs defined.

In pages 16-17: “To identify CCVs for each independent association signal, we conducted conditional analyses with adjustment of the lead variants for other signals in the same risk region. We conducted this analysis for trans-ancestral independent signals separately for each population to account for

differences in the LD structure and then combined conditioned results. Using the same approach described in a previous fine-mapping study for breast cancer¹², we defined a variant as CCV if it has a conditional P value within two orders of magnitude of the most significant association, conditioning on all other independent association signals.”

2. eQTL analyses – what tissue type was the transcriptome data derived from?

In page 18, paragraph 2: “We also conducted cis-expression quantitative trait loci (cis-eQTL) analyses to identify putative target genes using four transcriptome datasets derived from either normal colon tissues or tumor-adjacent normal colon tissues from 1,298 individuals from the Genotype-Tissue Expression (GTEx) project (n=368 individuals predominantly of European ancestry), the BarcUVa-Seq project (n=144 individuals of European ancestry), the Colonomics project (n=423 individuals of European ancestry), and the Asia Colorectal Cancer Consortium (ACCC) (n=363 individuals of East Asian ancestry) (**Methods**).”

3. What is meant by functional genomic evidence in Line 450 “Of these, 45 (21.9%) genes were supported by the functional genomic evidence.” Is this the target gene prediction from INQUISIT??

Yes, 45 genes identified in colocalization analysis of either mQTL or eQTL were predicted as targets of CCVs based on functional genomic data. We rephrased the description in the revised manuscript.

In page 19, paragraph 2: “Of these, 45 (21.9%) genes were also identified as targets of CCVs by in-silico analyses based on functional genomic data as described above, and 29 genes were identified in both mQTL and eQTL analyses.”

4. Line 470 – Please define ABS.

We included the definition of nine cell types in the single-cell RNA-seq data analysis in the Methods and Results sections.

In pages 35-36: “Nine cell types were defined: absorptive cells (ABS), adenoma-specific cells (ASC), crypt top colonocytes (CT), enteroendocrine cells (EE), goblet cells (GOB), stem cells (STM), serrated-specific cells (SSC), transit amplifying cells (TAC), and tuft cells (TUF). We identified differentially expressed genes (DEGs) by comparing each cell type with all other cell types and calculated a P-value for each gene using Wilcoxon's rank-sum test. The criteria $|\log_2 \text{fold change (FC)}| > 1$ and $P < 0.05$ were applied to determine genes with significantly differential expression between cell types.”

In page 20, paragraph 2: “Nine of these genes (DIP2B, CIB1, HPGD, CDKN2B, TMEM258, MYL12A, MYL12B, CDKN1A, and TMBIM1) showed a distinct expression pattern in specific absorptive cells (ABS), underscoring the relevance of this cell type underlying CRC development.”

5. In Figure 3 and 4. Please define which colocalization analyses were considered evidence.

We revised Figures 3 and 4 by indicating the type of colocalization analysis for each gene in the revised manuscript.

REVIEWERS' COMMENTS

Reviewer #1 (Remarks to the Author):

I acknowledge the authors' efforts in addressing my previous comments, which have contributed to a clearer manuscript. However, I still encounter challenges in understanding the column headings in the supplementary table, hindering the interpretation of the data.

(1) I kindly request a more detailed description of the Supplementary Tables to facilitate better comprehension.

(2) It would be beneficial to consolidate all potential functional effects of the CVV into a single table. While STable 20 appears to align with this suggestion, it remains unclear whether this table incorporates data from all other analyses. Clarification on this matter would enhance the overall transparency of the information presented.

Reviewer #2 (Remarks to the Author):

The authors have addressed all concerns adequately and have revised the manuscript accordingly.

Authors' Responses to Reviewers' Comments

Reviewer #1 (Remarks to the Author):

I acknowledge the authors' efforts in addressing my previous comments, which have contributed to a clearer manuscript. However, I still encounter challenges in understanding the column headings in the supplementary table, hindering the interpretation of the data.

(1) I kindly request a more detailed description of the Supplementary Tables to facilitate better comprehension.

Response: We appreciate the reviewer's valuable comments and have updated all Supplementary Tables with modified column headings and footnotes.

(2) It would be beneficial to consolidate all potential functional effects of the CVV into a single table. While STable 20 appears to align with this suggestion, it remains unclear whether this table incorporates data from all other analyses. Clarification on this matter would enhance the overall transparency of the information presented.

Response: The Supplementary Table 20 includes results from all analyses that we used to identify credible target genes. We have now made this clear in the revised Supplementary Table 20.